# The Impact of the Elemental Interactions on Soil Fertility and Toxicity in the Presence of Wastewater and Biosolids: A Quantitative Evaluation

Prodromos H. Koukoulakis [1], Panos Kanatas [2], Spyridon S. Kyritsis [2], Georgia Ntzala [2] and Ioannis K. Kalavrouziotis [2,*]

[1] Agricultural Research Organization DEMETRA, Soil Science Institute of Thessaloniki, 57001 Thermi, Greece
[2] School of Science and Technology, Hellenic Open University, 26335 Patras, Greece; kanatas.panagiotis@ac.eap.gr (P.K.)
[*] Correspondence: ikalabro@eap.gr

**Abstract:** A field experiment was conducted in Mesologgi, Greece, for the study of the elemental contribution to the soil under the following treatments in four replications i.e.,: a—Wastewater (TMWW), b—Biosolid (BSD), c—(TMWW + BSD) and d—CONTROL (fresh irrigation water). Similarly, the data of a greenhouse experiment conducted in four replications in Agrinion, under the effect of wastewater and biosolids was also taken into account for reasons of comparison. The soil analytical data of these two experiments were chosen to study the elemental interactions under two different experimental conditions The actual scope was the use of the elemental interactions as a tool for the evaluation of their contribution in terms of plant nutrients, and heavy metals to soil fertility and of heavy metals to soil toxicity. Based on the results of elemental contributions obtained for both of the above experimental soils, the key role of elemental interactions as a tool in evaluating the contributed heavy metals, and essential nutrients, as well as in producing quantitative changes in the physical and chemical properties of soil (pH, organic matter, calcium carbonate, and electrical conductivity), was also, studied. According to the obtained results, it was shown that the elemental interactions have shown approximately the same quantitative trend between some of the results obtained, differing in some others, showing higher concentrations. In other words, it was shown that the elemental interactions could be used as an effective tool for the quantitative evaluation of the elemental interactions' contribution in terms of nutrients to soil fertility, and of heavy metals to soil toxicity, under the reuse of wastewater and biosolids, as well as in terms of changes of the soil physical and chemical properties. However, due to the complex nature of this subject, more detailed research must be conducted on the elemental contributions, so that the plant nutrients, or the heavy metals, eventually be managed effectively to the benefit of the agricultural economy and environmental quality.

**Keywords:** elemental interactions contribution; heavy metals; regression analysis; regression equations; macroelements; microelements

## 1. Introduction

The presence of heavy metals in the environment constitutes a severe risk to human health and creates unfavorable conditions for plant growth, especially for the economic crops used for human nutrition [1,2]. However, these metals, in addition to their negative effects on the soil, plants, and the natural environment in general, can contribute positively to the improvement of soil fertility but also negatively generate soil toxicity, depending on their interactive orientation.

Synergistic elemental interactions between metals may create toxicity by contributing to the accumulation of metals in soil [3]. On the other hand, they may favor soil fertility if they interact synergistically with essential plant nutrients (macro and micronutrients),

consequently, supplying the soil with these elements. If the interactions with heavy metals are antagonistic, they may cause the reduction in heavy metals in the soil, thus favoring indirectly its fertility, because they contribute to the reduction in its toxicity, or if they act antagonistically with nutrients they may reduce the level of macro and micronutrients, thus affecting unfavorably soil fertility [4].

In conclusion, interactions play a dual role, i.e., they can both act favorably [5,6] or create problems in the soil [5,7,8]. In general, the effect of interactions in the soil depends on the type of interacting element, its concentration in the soil, the presence of other elements, soil moisture, temperature, soil type, metal binding capacity, and the presence of organic matter [9], clay content [10,11], etc.

In the past, reference was often made to the importance of interactions in general with respect to their contribution in terms of heavy metals and nutrients [12,13]. However, little has been said about the precise quantitative contribution of interactions in heavy metals, i.e., in relation to soil toxicity, as a result of their accumulation in soil, and also in relation to the interactions between heavy metals, nutrients and the physical and chemical properties of of soil, which affect the function of soil environment [14]. It is underlined that heavy metals do not only interact with each other, but also with macro micronutrients, as well as with the physical and chemical properties of the soil, and possibly with microorganisms [15] and generally with all factors of the biotic and abiotic system [16,17].

The role of interactions in the environment, as well as in the biotic and abiotic systems, is crucial. The interactions between various factors have been taking place endlessly since the genesis and appearance of the universe until today, controlling almost all manifestations and phenomena and the functions of physical, chemical, and biological processes occurring in nature. They have also been playing an essential role in plant animal and human life, in the macro and microcosmos to a greater or lesser extent. Consequently, due to their interactive contribution, they can be considered some of the main regulating agents of the quantified changes of all the environmental entities. This subject is undoubtedly a significant scientific and philosophical issue and goes beyond the actual scope of the present work.

Nevertheless, the elemental interactions are indeed significant, and exceptionally useful as in addition to their contribution of nutrients and metals to soils, they can also be applied as a method for cleaning wastewater or producing industrial products. They can also reveal information, for example, about the structural-functional relationships, as is the case of the interaction of Whey Protein Isolate (WPI) with Natural Deep Eutectic Solvents (NADESs). These interactions also help to determine the physical properties and develop dispersion between WPI and NADEs and also help to study the effects of NADEs on thermal and foam stability, surface tension, and conductivity [18]. Also, the metal and metalloids' mobility, bioavailability, and toxicity are influenced by their interactions with phyllosilicates, organic matter, charged minerals and microorganisms, sorption, desorption solution, complexation, oxidation-reduction, and precipitation–dissolution reactions [19]. All the above interrelations of interactions underline their importance.

Another favorable application of the interactions is that they can also be used as a method for the quantitative removal of toxic substances and qualitative improvement of wastewater produced by industry or households. This is possible by the removal of toxic elements or compounds with materials of high adsorbing capacity from the wastewater. For example, the wastewater of the textile and paper industry which contains the organic dye methyl orange, can successfully be removed by means of the interaction with chitosan-grafted nanocomposite, which is a strong adsorber of methyl orange. Such interactive methods are very popular in removing toxic components from wastewater, making possible their reuse after removing the toxic substances by interacting with strong organic adsorbers [20]. Recently, the currently conducted research is aiming towards developing and using adsorbing substances of heavy metals with the view to interact with wastewater, targeting the removal of the toxic metals of the polluted wastewater for improving their

quality. Some relevant examples of this type of research results based on such interactions are given below.

A number of papers related to the removal of heavy metals from wastewater have recently been published, such as on the interaction between a novel immobilized facial composite adsorbent for selecting Cu(II) ion detection and removal from wastewater [21]. In more recent work, the same author studied a ligand-supported mesoporous silica conjugate nanomaterial 0.0 for detecting and removing Cu(II) ions via adsorption [22]. Also, he used novel nanocomposite materials for efficient and selective mercury ions (Hg(II)) capturing from wastewater [23].

Similarly, the interaction of the ions of the chemical element Samarium (Sm(III)) (an element of the lanthanide series) is removed using adsorbed on 4-chlro-2-mercaptophenyl carbamodithioate (ACMPC) grafted onto mesoporous silica prepared for composite adsorption (CPA) on which the Samarium (III) ions are strongly adsorbed, and then chemically removed in pure form. This is a method of isolating Sm, which is a rare element [24].

In another publication, Awual [25] studied a particulate ligand anchored with highly ordered mesoporous silica-based nanocomposite material for detecting and adsorbing Cd(II) from wastewater. An eco-friendly process for toxic cadmium (II) removal by the interaction between the wastewater and the chemical ligand of 2,2′-Biquinoline-4,4′-dicarboxylic acid (BIDA) embedded on the mesoporous silica for the formation of facial composite adsorbent (FCA) studied by Sheikh et al. [26].

So far, the accurate quantitative contribution in terms of heavy metals by the elemental interactions to soil toxicity and fertility has not been evaluated. To the best of our knowledge, there has never been published relevant information in the bibliography about the use of elemental interactions as an effective method for the quantitative evaluation of the elemental interaction's contribution to soil fertility and toxicity, given that these interactions can change the soil chemical and physical properties, affecting, in the final analysis, the productive capacity of the soil.

Thus, the main objective of this paper is to provide experimental evidence about the elemental interactions occurring in the soil in order to be used as a method for the quantitative evaluation of the interactions' contribution to soil fertility and toxicity as a result of the accumulation in the soil of plant nutrients and heavy metals under the reuse of treated wastewater and biosolids.

Therefore, the present work aims to study the hundreds of elemental interactions occurring in the soil and use them as an effective method for the quantitative evaluation of the elemental interactive contribution to soil fertility and toxicity under the impact of the worldwide reuse of wastewater and biosolids.

## 2. Materials and Methods

A field experiment was established in Mesologgi Greece for the investigation of the effects of four wastewater and biosolid treatments, including:

1.   Treated Municipal Wastewater (TMWW).
2.   Biosolid (BSD).
3.   Treated Municipal Wastewater and Biosolid (TMWW + BSD).
4.   Fresh irrigation water (CONTROL).

The experiment included three replications with a total number of 12 experimental plots. A completely randomized block design was used, and the forage crop Fescue (*Festuca arundinacea* Schreb) was studied as a test crop. The plant biomass produced was cut at a mean 30cm plant height, and the dry matter yield of Fescue obtained is reported in Table 1.

**Table 1.** Dry matter yields of fescue plants (biomass) grown under the effect of wastewater and biosolids.

| Treatments | Replications (kg/ha) | | | Mean (kg/ha) |
|---|---|---|---|---|
| | I | II | III | |
| CONTROL | 30.00 | 29.00 | 31.00 | 30.00 |
| TMWW | 37.00 | 38.00 | 38.00 | 37.70 |
| TMWW + BSD | 82.00 | 80.00 | 79.00 | 80.33 |
| BSD | 62.00 | 67.00 | 71.00 | 67.70 |

### 2.1. Seedbed Preparation

The experimental seedbed preparation was conducted according to the following steps:

1.  The surface of the soil was cleaned so as to be firm and free of residues and weeds.
2.  The soil was plowed carefully, disced, and worked to fine tilth so as to come in close contact with the small fescue seeds as much as possible.
3.  The laying out of the experimental design on the soil surface was followed, and in turn, soil sampling was followed, taking one sample from each experimental plot, collecting in total 12 samples from a soil depth of 0–30cm.
4.  The seed was sown by uniform hand spreading on the surface of each plot, and then covering the seeds with a layer of soil 1–1.5 cm, followed by compacting it by means of a small roller so as to secure as much as possible better contact of the seeds with the soil, to facilitate effective seed germination.

### 2.2. Wastewater and Biosolids

*Wastewater:* The treated wastewater was supplied by the wastewater processing Center (WWPC) of Mesologgi, and its mean composition is shown in Table 2.

*Biosolids:* These biosolids were produced by draining the liquid fraction of sludge and followed by sun drying. The sludge was provided by the wastewater processing Center of Mesologgi. The composition of biosolids is reported in Table 3.

**Table 2.** The composition of wastewater applied to the fescue experiment.

| No | Elements | Concentration |
|---|---|---|
| 1 | pH | 7.70 |
| 2 | EC ($\mu$S/cm) (*) | 429.70 |
| 3 | COD (mg/L) (*) | 49.60 |
| 4 | TC (mg/L) (*) | 57.50 |
| 5 | TN (mg/L) (*) | 12.70 |
| 6 | $NH_4$-N (mg/L) | 12.60 |
| 7 | $NO_3$-N (mg/L) | 0.70 |
| 8 | $PO_4$-P (mg/L) | 3.94 |
| 9 | Ca (mg/L) | 2.84 |
| 10 | Mg (mg/L) | 26.77 |
| 11 | Fe (mg/L) | 32.50 |
| 12 | Mn (mg/L) | BDL (*) |
| 13 | Zn (mg/L) | 35.00 |
| 14 | Cu (mg/L) | 18.20 |
| 15 | Cd (mg/L) | BDL (*) |
| 16 | Co (mg/L) | 0.20 |
| 17 | Cr (mg/L) | BDL (*) |

Notes: (*) EC-Electrical conductivity, COD = Chemical Oxygen Demand, TC = total carbon. TN = Total nitrogen, BDL = Below Detection Limit.

**Table 3.** Composition of biosolids applied to the fescue experiment.

| No | Elements | Concentration |
|----|----------|---------------|
| 1 | $NO_3$ (mg/kg) | 166.30 |
| 2 | Fe (mg/kg) | 72.00 |
| 3 | Zn (mg/kg) | 141.0 |
| 4 | Mn (mg/kg) | 21.60 |
| 5 | Cu (mg/kg) | 20.00 |
| 6 | Cd (mg/kg) | BDL (*) |
| 7 | Co (mg/kg) | BDL (*) |
| 8 | Cr (mg/kg) | BDL (*) |
| 9 | Ca $CO_3$(%) | 5.55 |

Note: (*) BDL = Below Detection Limit.

## 2.3. Irrigation of Plants

The irrigation of fescue plants started at the appearance of two germinating leaves, and it was performed with the application of TMWW and fresh natural water according to the plan of the experimental design, i.e., the experimental plots 1, 2, 6, 8, 10, and 11 were irrigated, with a total of 450 L of TMWW per plot, while the plots 3, 4, 5, 7, 9, and 12 with 450 L per plot of fresh, natural water during the growth period.

## 2.4. Soil Analysis

The analysis of soil was made applying the following methods: Mechanical analysis by Bouyoucos [27] transformed by Gee and Bauer [28], Electrical Conductivity (EC) by Miller & Curtin [29], and the pH was measured on the saturated soil with the use of a standard electrode [30]. The $CaCO_3$ by the addition of 1N HCl, the excess of which was titrated with NaOH, while the percent of $CaCO_3$ was calculated by means of the relation:

$$CaCO_3(\%) = \frac{50}{10 \times 5} + \frac{x - y}{S} \tag{1}$$

where x = mL of NaOH for the titration of the control and y = mL for the titration of the excess HCl of soil [31,32]. The organic matter (OM) by the method of Walkley and Black (1934). The extent of the oxidation was measured by titration using a solution of $K_2Cr_2O_7$ in the presence of $H_2SO_4$.

On the other hand, the percent of the OM was calculated according to Nelson and Sommers [31], and Schumacher [32], the total organic C of soil being expressed in g, and the percent of organic matter was calculated by the relation [13]:

$$OM (\%) = organicC(g) \times 1.724 \times 1.33 \tag{2}$$

The results of soil analysis are reported in Table 4.

**Table 4.** Macro, micronutrients, and heavy metal content of the soil of fescue experiment.

| Elements | TMWW | | TMWW + BSD | | BSD | | CONTROL | |
|----------|------|--------|------------|--------|------|--------|---------|--------|
| | Mean | StdDev | Mean | StdDev | Mean | StdDev | Mean | StdDev |
| pH | 7.80 | 0.02 | 7.79 | 0.05 | 7.72 | 0.11 | 7.83 | 0.01 |
| EC (mS/cm) | 0.92 | 0.17 | 0.88 | 0.14 | 1.18 | 0.47 | 0.79 | 0.13 |
| OM (%) | 3.42 | 0.07 | 2.62 | 0.58 | 3.56 | 0.60 | 3.45 | 0.58 |
| $CaCO_3$ (%) | 8.77 | 4.21 | 4.83 | 6.33 | 3.33 | 2.02 | 5.13 | 5.30 |
| $NO_3$ (mg/kg) | 29.32 | 7.68 | 27.18 | 4.23 | 36.20 | 5.70 | 27.89 | 13.91 |
| $NO_3$-N (mg/kg) | 6.62 | 1.73 | 6.14 | 0.96 | 8.17 | 1.29 | 6.30 | 3.14 |
| P (mg/kg) | 18.22 | 1.51 | 15.33 | 1.14 | 25.59 | 10.67 | 16.65 | 2.20 |
| K (mg/kg) | 753.22 | 46.28 | 657.28 | 37.42 | 615.13 | 224.63 | 716.73 | 97.30 |
| Mg (mg/kg) | 561.26 | 49.72 | 502.10 | 29.67 | 479.49 | 146.84 | 525.17 | 80.63 |
| Ca (mg/kg) | 8574.43 | 307.50 | 7794.23 | 982.89 | 8442.61 | 1867.97 | 8626.04 | 701.70 |

**Table 4.** *Cont.*

| Elements | TMWW | | TMWW + BSD | | BSD | | CONTROL | |
|---|---|---|---|---|---|---|---|---|
| | Mean | StdDev | Mean | StdDev | Mean | StdDev | Mean | StdDev |
| Fe (mg/kg) | 32.84 | 2.00 | 25.59 | 9.24 | 30.44 | 15.67 | 25.09 | 9.92 |
| Zn (mg/kg) | 1.56 | 1.85 | 0.74 | 0.19 | 1.42 | 1.03 | 0.48 | 0.07 |
| Mn (mg/kg) | 6.96 | 0.55 | 6.98 | 0.82 | 7.35 | 0.82 | 6.37 | 0.47 |
| Cu (mg/kg) | 3.59 | 0.12 | 2.86 | 0.51 | 3.35 | 0.66 | 2.93 | 0.84 |
| B (mg/kg) | 1.41 | 0.52 | 1.32 | 0.33 | 1.48 | 0.65 | 1.22 | 0.39 |
| Na (mg/kg) | 226.26 | 51.55 | 164.12 | 19.01 | 176.22 | 75.16 | 189.22 | 89.64 |
| Cd (mg/kg) | 0.0303 | 0.0012 | 0.0217 | 0.0064 | 0.0270 | 0.0085 | 0.0250 | 0.0053 |
| Co (mg/kg) | 0.0230 | 0.0062 | 0.0193 | 0.0015 | 0.0203 | 0.0032 | 0.0177 | 0.0015 |
| Cr (mg/kg) | 0.0074 | 0.0005 | 0.0071 | 0.0000 | 0.0074 | 0.0005 | 0.0081 | 0.0017 |
| Ni (mg/kg) | 0.7217 | 0.0641 | 0.5517 | 0.1348 | 0.6733 | 0.2057 | 0.5990 | 0.1216 |
| Pb (mg/kg) | 1.8353 | 0.1903 | 1.4010 | 0.3312 | 1.5540 | 0.2998 | 1.4523 | 0.2812 |

As far as the available plant nutrients and heavy metals, they have been determined as follows: P by the method of Olsen [33], the exchangeable cations K, Ca, Mg, and Na by extraction with ammonium acetate, method of Lanyon and Heald [34], and the microelements and heavy metals were extracted with DTPA (diethylenetriaminepentaacetic acid) and were measured by ICP-OES Perkin Elmer.

*2.5. Statistical Analysis*

The statistical processing of the experimental data by means of the regression analysis was made by the use of the statistical package SPSS ver. 29. The simple regression analysis between two interacting variables was used for the study of the interactive contributions in metals and plant nutrients because this statistical procedure can give information about the contribution to soil of only one single element, which is the result of the interaction of only one element with another. On the other hand, the elemental contribution produced by the multiple regression is the product of the effect of a number of interacting elements. In working with elemental interaction, we are interested to know, for example, what is the interactive effect of P on Zn and vice versa so as to know how P affects its counterpart Zn.

## 3. Results and Discussion

*3.1. The Role of Interactions in the Ecosystem's Function between Minerals, Nutrients, and Soil Properties*

In this study, approximately more than 95% of the regression equations were found to be statistically significant, and their interactions contributed variable quantities of the following nutrients and non-nutrients and heavy metals: N, P, K, Ca, Mg, Na, Fe, Zn, Mn, Cu, Cd, Co, Cr, Ni, and Pb.

Determining the interactive contribution in terms of metals and nutrients by the regression equations was possible by solving separately each regression equation of the same group of equations mentioned below, i.e., from Equations (4)–(53). This solution was possible using the analytical, experimental data of soil. Note that each group of equations includes only the same dependent variable, as a function of one or two independent respective variables of each corresponding regression equation. The formation of the groups of equations having the same dependent variable depends on many factors, such as the type of the interacting element, its concentration, and the presence of other metals. For example, nitrates in the soil studied were found to be a function only of calcium carbonate, while P is a function of pH, Fe, Zn, Mn, and so on. This means that P interacted with all these independent variables and four regression equations, which, upon being solved, have more chances to contribute P synergistically or antagonistically to soil the final result being a balance between synergism versus antagonism. See Section 3.4 (ii) regression equations from Equations (5)–(9).

### 3.2. Evaluation of the Interactions Contribution to Metals and Nutrients

The total amount of macro or micronutrients or heavy metals corresponding to each group of the below-mentioned equations was calculated as follows:

1st: Each equation was solved using the experimental soil analytical data, so that the average value of the total contribution of each of the individual equations and each group, was calculated.

2nd: The above mean value obtained, was considered the contributed "Mean Interactive Value" of the metal or nutrient in question.

3rd: The average mean soil concentration of each metal or macro and micronutrient, respectively, given by the soil analysis, was taken into account, and

4th: The following relationship (3) was used to calculate the *Pic* (Percent Interactive Contribution).

$$Pic = \frac{(Mic - Msec) \times 100}{Mic} \qquad (3)$$

where:

*Pic* = Percent Interactive Contribution

*Mic* = Mean Interactive contribution

*Msec* = Mean soil elemental content

The elemental interactive contribution varies not only with the chemical affinity of the element supplied but also with the changes in physical and chemical properties of the soil, as well as with the general conditions prevailing in the soil. The *Pic* can be positive or negative, the latter meaning that the mean interactive contribution (*Mic*) is smaller than the mean soil elemental contribution (*Msec*), thus yielding a negative "percent interactive contribution" (*Pic*).

### 3.3. Contribution of Interactions in Macro and Micronutrients

The above procedure (method) for calculating the interactive contribution is made possible by regression analysis of the soil analytical data of both the Fescue and the Lettuce experiments, respectively. The relevant results for Fescue soil are reported in Table 5 below and were calculated based on the following regression equations in Section 3.4 from Equations (4)–(53). Also, the results of the lettuce soil are shown in Table 6.

**Table 5.** Contribution by the interactions between metals, nutrients, physical and chemical properties of soil in plant nutrients under the influence of treated wastewater and biosolids and under the cultivation of *Festuca arundinacea* Schreb.

| # | Macro and Micronutrients Interacting in Soil | Average Interactive Contribution in Macro and Micronutrients (mg/kg) (a) | Average Value in Soil Macro and Micronutrients (mg/kg) (b) | Contribution in Interactive Macro and Micronutrients (mg/kg) (c) | Percent Interactive Contribution (%) (d) |
|---|---|---|---|---|---|
| 1 | NO$_3$ | 30.15 | 30.15 | 0.00 | 0.00 |
| 2 | P | 14.33 | 18.98 | −0.200 | −1.39 |
| 3 | K | 686.39 | 685.59 | 0.80 | +0.12 |
| 4 | Mg | 378.33 | 517.00 | −138.67 | −36.65 |
| 5 | Ca | 8359.22 | 8359.33 | −0.11 | −0.001 |
| 6 | Cu | 2.32 | 3.19 | −0.87 | −37.50 |
| 7 | Mn | 7.00 | 6.92 | 0.088 | +1.26 |
| 8 | Zn | 1.05 | 1.05 | 0.00 | 0.00 |
| 9 | B | 1.90 | 1.36 | 0.54 | +28.42 |

The interactive contribution of macro and micronutrients to soil is clearly presented in Table 5, and the following information is given for a better understanding of this Table.

For each macro or micronutrient, four values correspond horizontally, i.e., the first value (a) refers to the mean interactive element, contributed by the following equations in Section 3.4 Equations (4)–(53). The second (b) value refers to the average concentration of the element in soil, which is also due, among other factors, to interactions. The third value (c), is the difference between the first and the second, i.e., (a)–(b), which constitutes the basic contribution of the interactions of the element under consideration. Finally, the fourth (d) value is the percent (%) elemental contribution by the interaction in macro and micronutrient, calculated by the above relationship (3).

For example, the interactive contribution in B due to interactions is 1.90 mg/kg. The mean value of B in the experimental soil is 1.36 mg/kg, while the difference between these values is 0.54 mg/kg, and the percent interactive contribution in B content is 28.42% mg/kg. Based on the soil sampling parameters, i.e., soil depth 0–30 cm, volume weight 1.35 g/cm$^3$, the soil mass corresponding to this depth is $0.30 \times 1.35 \times 10^7 = 4{,}050{,}000$ kg soil/ha. Therefore, the total B contributed by the interactions/ha is $(4{,}050{,}000/10^6) \times 0.54 = 2.2$ kg B/ha and in terms of boron fertilizer (borax), is equal to $2.2 \times 100/14.5 = 15.2$ kg of Borax fertilizer per ha. This is equivalent to the positive contribution to the soil of the elemental interactions in terms of B.

Also, the contribution in terms of Mn is equal to $(4{,}050{,}000/10^6) \times 0.088 = 0.356$ kg Mn/ha, and in terms of MnSO$_4$ fertilizer with 32% Mn content, is equal to $(0.356 \times 100)/32 = 1.11$ kg MnSO$_4$/ha.

Regarding the contribution in terms of the other nutrients, such as Ca, Mg, and Zn, (Table 5), they are either negative or zero, meaning that the interactions were mainly antagonistic.

It is emphasized here that in general, based on the results of Table 5, it is concluded that the outcome (contribution) of the elemental interaction is neither static nor indefinite because the interactions take place continuously and endlessly, and therefore, the concentrations of the interactively contributed elements change also continuously. However, these results show that the interactions play an important role in maintaining soil fertility and perpetually, inducing quantitative changes in soil nutrient levels, and invariably affecting plant growth and yields. For all these reasons, it is underlined that the interactions are a dynamic process, and their quantitative contribution is never constant. It varies over time depending on the prevailing conditions, and this reality complicates their study. Many factors, known and unknown, affect their effectiveness. This is also evident in Tables 5–10 data. Thus, the contribution may be zero or negative, or conversely, positive due to the above reasons.

A key factor affecting the functions of the interactions is the number of interactive elements with which the dependent variable interacts. A given independent variable can interact with one or two or more variables (elements).

The study in Table 5 shows that the soil is positively supplied with interactive B (+28.42), Mn (+1.26), & K (+0.12%), negatively with Mg (−36.65%), Ca (−0.001%), and Cu (−37.50%) and finally with zero NO$_3$ & Zn (0.00%). The later results, of course, reflect the occurrence of antagonistic interactions prevailing during soil sampling. We do not know whether these data will change soon in the experimental soil from which the samples were taken. However, by continuing the relevant research, hopefully, we could have approximate mean information about the overall changes in metal concentrations so as to understand better the interactive evolution and its elemental contribution in space and time.

In conclusion, it can be supported that the elemental interactions have played an important role since the creation of the universe, that is, millions of years ago, and continue to control the level of concentration of nutrients and non-nutrients and determine to a great extent the fertility of the soil and consequently the growth of plants, as well as the level of soil toxicity according to the degree of accumulation of heavy metals in the soil. Continuous research effort is required to understand better the role of interactions in soil, plants, and the environment in general in order to acquire practical benefits, such as exploiting the role of the contribution of interactions in nutrients for more economical and friendly fertilization

of crops, but also for more effectively addressing the problems of soil and the environment, in general.

*3.4. Interactive Regression Equations between Heavy Metals and Nutrients in the Experimental Soil of Festuca arundinacea Schreb*

(i)    Regression equations of Nitrogen contribution (N)

$$NO_3 = -1.147 \times CaCO_3 + 36.474$$
$$(R^2 = 0.40 \quad sig = 0.029 \quad N = 12)$$

(4)

(ii)    Regression equations of phosphorus contribution (P)

$$P = 0.210 \times (Sludge)^2 - 17.675 \times (Sludge) + 387.544$$
$$(R^2 = 0.66 \quad sig = 0.008 \quad N = 12)$$

(5)

$$P = -80.389 \times pH + 644.709$$
$$(R^2 = 0.70 \quad sig = 0.001 \quad N = 12)$$

(6)

$$P = 0.052 \times Fe^2 - 3.082 \times Fe + 59.850$$
$$(R^2 = 0.53 \quad sig = 0.0034 \quad N = 12)$$

(7)

$$P = -4.902 \times Zn^2 + 22.433 \times Zn + 5.516$$
$$(R^2 = 0.75 \quad sig = 0.002 \quad N = 12)$$

(8)

$$P = 4.904 \times Mn^2 - 62.902 \times Mn + 217.283$$
$$(R^2 = 0.53 \quad sig = 0.033 \quad N = 12)$$

(9)

(iii)    Regression equations of potassium, contribution (K)

$$K = -0.619 \times C^2 + 62.688 \times C - 782.388$$
$$(R^2 = 0.54 \quad sig = 0.029 \quad N = 12)$$

(10)

$$K = -1.482 \times (Sludge)^2 + 104.183 \times (Sludge) - 1031.05$$
$$(R^2 = 0.60 \quad sig = 0.01 \quad N = 12)$$

(11)

$$K = -0.001 \times Mg^2 + 2.373 \times Mg - 281.688$$
$$(R^2 = 0.96 \quad sig < 0.001 \quad N = 12)$$

(12)

$$K = -162.488 \times Mn^2 + 2205.796 \times Mn - 6727.661$$
$$(R^2 = 0.59 \quad sig = 0.019 \quad N = 12)$$

(13)

$$K = 167.573 \times Cu^2 - 943.089 \times Cu + 1935.151$$
$$(R^2 = 0.64 \quad sig = 0.032 \quad N = 12)$$

(14)

$$K = -0.012 \times Na^2 + 6.742 \times Na - 112.325$$
$$(R^2 = 0.85 \quad sig < 0.001 \quad N = 12)$$

(15)

$$K = 15072.82 \times Cd + 293.699$$
$$(R^2 = 0.57 \quad sig = 0.004 \quad N = 12)$$

(16)

$$K = -382.063 \times Pb^2 + 1466.164 \times Pb - 641.199$$
$$(R^2 = 0.41 \quad sig = 0.045 \quad N = 12)$$

(17)

(iv)    Regression equations of Ca contribution (Ca)

$$Ca = -16.829 \times (Sludge)^2 + 1277.613 \times (Sludge) - 15198.023$$
$$(R^2 = 0.50 \quad sig = 0.047 \quad N = 12)$$

(18)

$$Ca = -3.795 \times Fe^2 + 296.667 \times Fe + 3302.157$$
$$(R^2 = 0.57 \quad sig = 0.01 \quad N = 12)$$
(19)

$$Ca = -798.905 \times Mn^2 + 10130.539 \times Mn - 23146.783$$
$$(R^2 = 0.63 \quad sig = 0.010 \quad N = 12)$$
(20)

$$Ca = 9138192.614 \times Cd^2 + 586258.369 \times Cd - 396.790$$
$$(R^2 = 0.54 \quad sig = 0.035 \quad N = 12)$$
(21)

$$Ca = -6410.483 \times Pb^2 + 2251.102 \times Pb - 10080.202$$
$$(R^2 = 0.54 \quad sig = 0.031 \quad N = 12)$$
(22)

(v)  Regression equations of Magnesium contribution (Mg)

$$Mg = -0.398 \, C^2 + 40.867 \times C - 447.433$$
$$(R^2 = 0.55 \quad sig = 0.029 \quad N = 12)$$
(23)

$$Mg = -0.995 \times Sludge^2 + 69.833 \times Sludge - 629.86$$
$$(R^2 = 0.60 \quad sig = 0.016 \quad N = 12)$$
(24)

$$Mg = -0.205 \times Fe^2 + 19.035 \times Fe + 157.925$$
$$(R^2 = 0.70 \quad sig = 0.004 \quad N = 12)$$
(25)

$$Mg = -124.017 \times Mn^2 + 1701.816 \times Mn - 5267.206$$
$$(R^2 = 0.64 \quad sig = 0.010 \quad N = 12)$$
(26)

$$Mg = 84.647 \times Cu^2 - 448.969 \times Cu + 1060.961$$
$$(R^2 = 0.52 \quad sig = 0.036 \quad N = 12)$$
(27)

$$Mg = 10173.167 \times Cd + 252.504$$
$$(R^2 = 0.57 \quad sig = 0.005 \quad N = 12)$$
(28)

$$Mg = -400.552 \times Pb^2 + 1439.213 \times Pb - 721.202$$
$$(R^2 = 0.50 \quad sig = 0.003 \quad N = 12)$$
(29)

(vi)  Regression equations of Sodium contribution (Na)

$$Na = 0.002 \times Mg^2 - 1.435 \times Mg + 343.474$$
$$(R^2 = 0.92 \quad sig < 0.001 \quad N = 12)$$
(30)

$$Na = -0.199 \times Fe^2 + 11.954 \times Fe - 44.815$$
$$(R^2 = 0.63 \quad sig = 0.011 \quad N = 12)$$
(31)

$$Na = 34.840 \times Cu^2 - 142.845 \times Cu + 279.208$$
$$(R^2 = 0.65 \quad sig = 0.010 \quad N = 12)$$
(32)

$$Na = 65325.22 \times Cd^2 + 4297.963 \times Cd + 30.842$$
$$(R^2 = 0.63 \quad sig = 0.015 \quad N = 12)$$
(33)

$$Na = -707.677 \times Pb^2 + 785.013 \times Pb - 526.185$$
$$(R^2 = 0.61 \quad sig = 0.015 \quad N = 12)$$
(34)

(vii)  Regression equations of iron contribution (Fe)

$$Fe = 1.095 \times C - 13.314$$
$$(R^2 = 0.55 \quad sig = 0.006 \quad N = 12)$$
(35)

$$\text{Fe} = -0.010 \times (\text{sludge})^2 - 0.955 \times (\text{sludge}) + 88.991$$
$$(R^2 = 0.64 \quad \text{sig} = 0.010 \quad N = 12) \tag{36}$$

$$\text{Fe} = 0.008 \times \text{Mg}^2 - 0.058 \times \text{Mg} + 16.215$$
$$(R^2 = 0.67 \quad \text{sig} = 0.006 \quad N = 12) \tag{37}$$

$$\text{Fe} = 5.118 \times \text{Cu}^2 - 18.256 \times \text{Cu} + 33.066$$
$$(R^2 = 0.84 \quad \text{sig} = 0.001 \quad N = 12) \tag{38}$$

$$\text{Fe} = -0.001 \times \text{Na}^2 + 0.628 \times \text{Na} - 42.155$$
$$(R^2 = 0.78 \quad \text{sig} = 0.001 \quad N = 12) \tag{39}$$

$$\text{Fe} = 56983.232 \times \text{Cd}^2 - 1471.543 \times \text{Cd} + 26.301$$
$$(R^2 = 0.92 \quad \text{sig} < 0.001 \quad N = 12) \tag{40}$$

$$\text{Fe} = -36.730 \times \text{Pb}^2 + 142.396 \times \text{Pb} - 101.318$$
$$(R^2 = 0.81 \quad \text{sig} < 0.001 \quad N = 12) \tag{41}$$

(viii) Regression equations of Zinc contribution (Zn)

$$\text{Zn} = 13.592 \times \text{Pb}^2 - 41.514 \times \text{Pb} + 31.635$$
$$(R^2 = 0.72 \quad \text{sig} = 0.0013 \quad N = 12) \tag{42}$$

(ix)   Copper contribution equations (Cu)

$$\text{Cu} = 2.32 \times \text{C} - 5.238$$
$$(R^2 = 0.41 \quad \text{sig} = 0.025 \quad N = 12) \tag{43}$$

$$\text{Cu} = -0.088 \times \text{Sludge} + 6.953$$
$$(R^2 = 0.37 \quad \text{sig} = 0.036 \quad N = 12) \tag{44}$$

$$\text{Cu} = 2.964 \times 10^{-5} \times \text{Mg}^2 - 0.024 \times \text{Mg} + 7.582$$
$$(R^2 = 0.66 \quad \text{sig} = 0.008 \quad N = 12) \tag{45}$$

$$\text{Cu} = 0.057 \times \text{Fe} + 1.575$$
$$(R^2 = 0.81 \quad \text{sig} < 0.001 \quad N = 12) \tag{46}$$

$$\text{Cu} = -2.348 \times 10^{-5} \times \text{Na}^2 + 0.017 \times \text{Na} + 0.810$$
$$(R^2 = 0.64 \quad \text{sig} = 0.012 \quad N = 12) \tag{47}$$

$$\text{Cu} = 40419.448 \times \text{Cd}^2 - 119.448 \times \text{Cd} + 3.437$$
$$(R^2 = 0.85 \quad \text{sig} < 0.001 \quad N = 12) \tag{48}$$

$$\text{Cu} = -0.807 \times \text{Pb}^2 + 4.267 \times \text{Pb} - 1.444$$
$$(R^2 = 0.77 \quad \text{sig} = 0.001 \quad N = 12) \tag{49}$$

(x)   Regression equations of Manganese contribution (Mn)

$$\text{Mn} = -7.063 \times \text{pH} + 61.893$$
$$(R^2 = 0.45 \quad \text{sig} = 0.0017 \quad N = 12) \tag{50}$$

$$\text{Mn} = 1.315 \times \text{B} + 5.131$$
$$(R^2 = 0.68 \quad \text{sig} = 0.001 \quad N = 12) \tag{51}$$

(xi)   Regression equations of Boron contribution (B)

$$\text{B} = -4.549 \times \text{pH} + 36.771$$
$$(R^2 = 0.48 \quad \text{sig} = 0.0013 \quad N = 12) \tag{52}$$

$$B = 3.537 \times 10^{-5} \times Mg^2 - 0.034 \times Mg + 9.165$$
$$(R^2 = 0.62 \quad sig = 0.001 \quad N = 12)$$

(53)

The study of the above regression equations from Equations (4)–(53) shows that each group of equations includes the same dependent variable, which within the same group is a function of various metals, nutrients, or physical and chemical properties of the soil as shown in each respective regression equation. Each equation within the same group contributes a corresponding amount of the dependent variable according to its interactive orientation (synergistic or antagonistic) and consequently affects the dependent variable and, hence, the soil conditions.

*3.5. Contribution of Interactions in Macro and Micronutrients in the Experimental Soil of Lactuca sativa var. Longifolia (1st Soil Sampling)*

In order to compare the data of the interactive contribution given in Table 5 for the Fescue experiment with similar data obtained by means of other experiments, the relevant research of Ntzala [35], who conducted an experiment applying treated wastewater and biosolid to lettuce, was used. From the regression analysis of the soil analytical data of this experiment, the relevant contribution results are reported by solving equations from Equations (54)–(120).

**Table 6.** Contribution in terms of essential plant nutrients and heavy metals by the interactions between metals, nutrients, and chemical properties of soil under the influence of treated wastewater and biosolids in the presence of *Lactuca sativa* var. *Longifolia* [35].

| # | Macro and Micronutrients Interacting in Soil | Average Interactive Contribution to Macro and Micronutrients (mg/kg) | Average Value in Soil Macro and Micronutrients (mg/kg) | Contribution in Interactive Macro and Micronutrients (mg/kg) | Percent Interactive Contribution (%) |
|---|---|---|---|---|---|
| 1 | N (%) | 0.132 | 0.132 | 0.00 | 0.00 |
| 2 | P(mg/kg) | 100.22 | 101.71 | 2.49 | −1.49 |
| 3 | K(mg/kg) | 124.79 | 128.36 | −2.86 | −2.29 |
| 4 | Ca(mg/kg) | 207.84 | 206.02 | 0.12 | +0.88 |
| 5 | Mg(mg/kg) | 31.54 | 30.18 | 1.36 | +4.31 |
| 6 | Zn(mg/kg) | 2.29 | 2.24 | 0.05 | +2.18 |
| 7 | Fe(mg/kg) | 20.63 | 22.77 | −2.14 | −10.37 |
| 8 | Mn(mg/kg) | 33.59 | 33.50 | 0.09 | +0.27 |
| 9 | Cu(mg/kg) | 2.50 | 2.10 | 0.40 | +16.00 |

This contribution, as given in Table 6, follows the same trend as the data in Table 5. Naturally, there are some differences in elemental contribution between the macro or micronutrients compared to those obtained with the Fescue experiment. E.g., the contribution of Zn in the soil of the fescue experiment was 0% and in Cu −37.50% (Table 5), while in the case of lettuce, Zn +2.18% and Cu +16% (Table 6). These differences are to be expected and are due to the dynamic nature of the interactions. However, the fact that must be emphasized is that the interactions can contribute positively or negatively at a high or low percent level. The increase or decrease in soil fertility through the nutrient concentration changes caused by the elemental interactions, regardless of time and space, undoubtedly underlines the critical role of interactions in regulating the level of soil nutrients and soil fertility.

The data of Tables 5 and 6 are compared in relation to the interactive contribution of essential nutrients expressed in terms of actual fertilizers. The results obtained are shown in Table 7 as follows:

**Table 7.** Interactive contribution to soil of the Fescue and Lettuce experiment under the effect of treated wastewater and biosolids.

| Experimental Soil | Elements Contributed | mg/kg Soil | Contributed Fertilizer | kg/ha |
|---|---|---|---|---|
| Fescue experiment | K | 0.80 | Potassium sulfate (50% $K_2O$) | 7.80 |
| | Mn | 0.09 | Manganese sulfate (35% Mn) | 1.00 |
| | B | 0.54 | Borax (11.5% B) | 19.60 |
| | P | 2.49 | 0-46-0 (N-$P_2O_5$-$K_2O$) | 57.60 |
| | Ca | 0.12 | $CaCO_3$ | 1.40 |
| Lettuce experiment | Mg | 1.36 | $MgSO_2$ (25% Mg) | 31.3 |
| | Zn | 0.05 | $ZnSO_4.7H_2O$ (25% Zn) | 0.50 |
| | Mn | 0.02 | $MnSO_4$ (35% Mn) | 0.29 |
| | Cu | 0.40 | $CuSO_4$ (25% Cu) | 1.86 |

As it is seen, the study of the data in Table 7 discloses that there are some basic quantitative and qualitative differences between the elemental interactive contribution in terms of the kind of elements between the Fescue and lettuce soil, i.e., in the number of contributed elements K, Mn, and B in Fescue soil, and P, Ca, Mg, Zn, Mn, and Cu in lettuce soil. Similarly, there is a significant difference between the elements of the same soil and in both soils. Also, the same element in both soils, such as Mn, has been contributed at a higher level in the fescue soil than in the lettuce soil. All these differences are related to various known and possibly unknown factors, such as the concentration and the presence of the kind of metals, macro, and micronutrients, their interactive capacity, the soil properties such as pH, organic matter, clay content, electrical conductivity, soil microorganisms, mobility of elements in the soil, and many other factors, such as the extent of fixation, soil moisture, and soil temperature. Generally, the behavior of the elements is directly related to the soil properties. The factors that affect the behavior of nutrients and heavy metals are pH, organic matter, electrical conductivity, oxidation-reduction potential, and oxides of Fe and Mn [36]. The above factors affect the adsorption, resorption, and mobility of metals and nutrients, a fact that is directly related to their availability and, hence, their interactive behavior.

It is necessary that future research must be conducted for a number of years with the basic economic crops so as to find an average "percent interactive contribution" for each plant nutrient to be taken into account in the fertilizer management context.

*3.6. Regression Equations and the Contribution of Their Interactions in Macro and Micronutrients to the Soil of Lettuce (Lactuca sativa var. Longifolia) (1st Soil Sampling)*

(i)　Regression equations of Nitrogen contribution (N)

$$N = 0.003 \times Mg + 0.041$$
$$(R^2 = 0.21 \quad sig < 0.001 \quad N = 48) \tag{54}$$

$$N = 0.478 \times (OM)^2 - 1.614 \times (OM) + 1.472$$
$$(R^2 = 0.17 \quad sig < 0.013 \quad N = 48) \tag{55}$$

(ii)　Regression equations of phosphorus contribution (P)

$$P = -883.299 \times N^2 + 802.343 \times N + 20.473$$
$$(R^2 = 0.48 \quad sig < 0.001 \quad N = 48) \tag{56}$$

$$P = -2.004 \times pH^2 + 19.679 \times pH + 57.2136$$
$$(R^2 = 0.29 \quad sig < 0.001 \quad N = 48) \tag{57}$$

$$P = -0.005 \times K^2 + 1.442 \times K + 2.465$$
$$(R^2 = 0.59 \quad sig < 0.001 \quad N = 48) \tag{58}$$

$$P = -0.003 \times Ca^2 + 1.078 \times Ca - 0.069$$
$$(R^2 = 0.77 \quad sig < 0.001 \quad N = 48)$$
(59)

$$P = \ln(Mg) \times 13.448 + 58.863$$
$$(R^2 = 0.68 \quad sig < 0.001 \quad N = 48)$$
(60)

$$P = -0.144 \times Fe^2 + 7.88 \times Fe - 0.920$$
$$(R^2 = 0.81 \quad sig < 0.001 \quad N = 48)$$
(61)

$$P = -2.537 \times Zn^2 + 21.719 \times Zn + 69.574$$
$$(R^2 = 0.40 \quad sig < 0.001 \quad N = 48)$$
(62)

$$P = -0.078 \times Mn^2 + 5.654 \times Mn + 3.72$$
$$(R^2 = 0.75 \quad sig < 0.001 \quad N = 48)$$
(63)

(iii)　Regression equations of potassium contribution (K)

$$K = 0.682 \times pH^2 + 5.317 \times pH + 75.780$$
$$(R^2 = 0.16 \quad sig = 0.021 \quad N = 48)$$
(64)

$$K = -1079.583 \times N^2 + 986.319 \times N + 28.313$$
$$(R^2 = 0.47 \quad sig < 0.001 \quad N = 48)$$
(65)

$$K = -0.003 \times Ca^2 + 1.262 \times Ca - 1.120$$
$$(R^2 = 0.45 \quad sig < 0.001 \quad N = 48)$$
(66)

$$K = \ln(Mg) \times 15.512 + 79.042$$
$$(R^2 = 0.33 \quad sig < 0.001 \quad N = 48)$$
(67)

$$K = -0.193 \times Fe^2 + 10.165 \times Fe + 0.614$$
$$(R^2 = 0.45 \quad sig < 0.001 \quad N = 48)$$
(68)

$$K = \ln(Zn) \times 21.841 + 113.986$$
$$(R^2 = 0.27 \quad sig < 0.001 \quad N = 48)$$
(69)

$$K = -0.127 \times Mn^2 + 8.003 \times Mn + 9.252$$
$$(R^2 = 0.48 \quad sig < 0.001 \quad N = 48)$$
(70)

(iv)　Regression equations of calcium contribution (Ca)

$$Ca = -2.739 \times pH^2 + 33.461 \times pH + 108.234$$
$$(R^2 = 0.10 \quad sig < 0.001 \quad N = 48)$$
(71)

$$Ca = -942.383 \times N^2 + 850.488 \times N + 120.075$$
$$(R^2 = 0.13 \quad sig = 0.048 \quad N = 48)$$
(72)

$$Ca = -0.024 \times P^2 + 4.537 \times P + 0.024$$
$$(R^2 = 0.53 \quad sig < 0.001 \quad N = 48)$$
(73)

$$Ca = \ln(K) \times 25.773 + 84.991$$
$$(R^2 = 0.50 \quad sig < 0.001 \quad N = 48)$$
(74)

$$Ca = \ln(Mg) \times 22.189 + 135.483$$
$$(R^2 = 0.43 \quad sig < 0.001 \quad N = 48)$$
(75)

$$Ca = 33.258 \times \ln(Fe) + 105.896$$
$$(R^2 = 0.50 \quad sig < 0.001 \quad N = 48)$$
(76)

$$Ca = \ln(Zn) \times 38.408 + 180.754$$
$$(R^2 = 0.29 \quad sig < 0.001 \quad N = 48)$$
(77)

$$Ca = -0.133 \times Mn^2 + 10.185 \times Mn + 21.364$$
$$(R^2 = 0.62 \quad sig < 0.001 \quad N = 48)$$
(78)

(v) Regression equations of Magnesium contribution (Mg)

$$Mg = -111.810 \times N^2 + 173.979 \times N + 10.948$$
$$(R^2 = 0.12 \quad sig = 0.003 \quad N = 48)$$
(79)

$$Mg = 0.279 \times P + 1.767$$
$$(R^2 = 0.84 \quad sig = 0.037 \quad N = 48)$$
(80)

$$Mg = -0.002 \times Ca^2 + 0.584 \times Ca + 1.759$$
$$(R^2 = 0.21 \quad sig = 0.006 \quad N = 48)$$
(81)

$$Mg = 1.043 \times Mn - 4.549$$
$$(R^2 = 0.22 \quad sig < 0.001 \quad N = 48)$$
(82)

$$Mg = 85081.738 \times Co^2 - 3805.239 \times Co + 69.145$$
$$(R^2 = 0.15 \quad sig = 0.0030 \quad N = 48)$$
(83)

$$Mg = -1.181 \times Pb^2 + 14.220 \times Pb + 2.106$$
$$(R^2 = 0.50 \quad sig = 0.020 \quad N = 48)$$
(84)

(vi) Regression equations of Iron contribution (Fe)

$$Fe = -0.691 \times pH^2 + 6.120 \times pH + 11.176$$
$$(R^2 = 0.32 \quad sig < 0.001 \quad N = 48)$$
(85)

$$Fe = -179.980 \times N^2 + 163.477 \times N + 6.220$$
$$(R^2 = 0.33 \quad sig < 0.001 \quad N = 48)$$
(86)

$$Fe = -0.001 \times P^2 + 0.332 \times P - 0.072$$
$$(R^2 = 0.69 \quad sig < 0.001 \quad N = 48)$$
(87)

$$Fe = -0.001 \times K^2 + 0.327 \times K + 0.280$$
$$(R^2 = 0.65 \quad sig < 0.001 \quad N = 48)$$
(88)

$$Fe = \ln(Mn) \times 2.845 + 13.728$$
$$(R^2 = 0.50 \quad sig < 0.001 \quad N = 48)$$
(89)

$$Fe = \ln(Zn) \times 4.688 + 19.688$$
$$(R^2 = 0.57 \quad sig = 0.001 \quad N = 48)$$
(90)

$$Fe = -0.013 \times Mn^2 + 1.132 \times Mn + 0.400$$
$$(R^2 = 0.67 \quad sig = 0.001 \quad N = 48)$$
(91)

$$Fe = 120.14 \times Ni^2 - 8.157 \times Ni + 15.379$$
$$(R^2 = 0.34 \quad sig < 0.001 \quad N = 48)$$
(92)

(vii) Regression equations of Zinc contribution (Zn)

$$Zn = 0.001 \times EC^2 - 0.051 \times EC + 2.173$$
$$(R^2 = 0.43 \quad sig < 0.001 \quad N = 48)$$
(93)

$$\text{Zn} = -0.001 \times \text{Mg}^2 + 0.100 \times \text{Mg} + 0.671$$
$$(\text{R2} = 0.13 \quad \text{sig} = 0.041 \quad \text{N} = 48)$$

(94)

$$\text{Zn} = 0.006 \times \text{Fe}^2 - 0.065 \times \text{Fe} + 0.579$$
$$(\text{R}^2 = 0.22 \quad \text{sig} = 0.003 \quad \text{N} = 48)$$

(95)

$$\text{Zn} = 0.063 \times \text{Mn} + 0.132$$
$$(\text{R2} = 0.14 \quad \text{sig} = 0.010 \quad \text{N} = 48)$$

(96)

$$\text{Zn} = -1.199 \times \text{Cu}^2 + 7.191 \times \text{Cu} - 7.305$$
$$(\text{R}^2 = 0.23 \quad \text{sig} = 0.003 \quad \text{N} = 48)$$

(97)

$$\text{Zn} = 50.753 \times \text{Cr} - 0.293$$
$$(\text{R}^2 = 0.10 \quad \text{sig} = 0.030 \quad \text{N} = 48)$$

(98)

$$\text{Zn} = 184.899 \times \text{Ni}^2 - 93.098 \times \text{Ni} + 13.460$$
$$(\text{R}^2 = 0.23 \quad \text{sig} = 0.003 \quad \text{N} = 48)$$

(99)

(viii) Regression equations of Manganese contribution (Mn)

$$\text{Mn} = -1.195 \times \text{pH}^2 + 9.972 \times \text{pH} + 16.910$$
$$(\text{R}^2 = 0.23 \quad \text{sig} = 0.003 \quad \text{N} = 48)$$

(100)

$$\text{Mn} = -0.003 \times \text{P}^2 + 0.787 \times \text{P} + 0.526$$
$$(\text{R}^2 = 0.46 \quad \text{sig} < 0.001 \quad \text{N} = 48)$$

(101)

$$\text{Mn} = -0.002 \times \text{K}^2 + 0.535 \times \text{K} + 3.223$$
$$(\text{R}^2 = 0.48 \quad \text{sig} < 0.001 \quad \text{N} = 48)$$

(102)

$$\text{Mn} = -0.001 \times \text{Ca}^2 + 0.314 \times \text{Ca} + 0.836$$
$$(\text{R}^2 = 0.45 \quad \text{sig} < 0.001 \quad \text{N} = 48)$$

(103)

$$\text{Mn} = \ln(\text{Mg}) \times 5.106 + 17.264$$
$$(\text{R}^2 = 0.55 \quad \text{sig} < 0.001 \quad \text{N} = 48)$$

(104)

$$\text{Mn} = -0.034 \times \text{Fe}^2 + 2.264 \times \text{Fe} + 0.362$$
$$(\text{R}^2 = 0.52 \quad \text{sig} < 0.001 \quad \text{N} = 48)$$

(105)

$$\text{Mn} = \ln(\text{Zn}) \times 7.321 + 28.679$$
$$(\text{R}^2 = 0.48 \quad \text{sig} < 0.001 \quad \text{N} = 48)$$

(106)

$$\text{Mn} = 3.735 \times \text{Cu}^2 - 13.842 \times \text{Cu} + 45.599$$
$$(\text{R}^2 = 0.14 \quad \text{sig} < 0.031 \quad \text{N} = 48)$$

(107)

$$\text{Mn} = 20273.705 \times \text{Cr}^2 - 1714.287 \times \text{Cr} + 67.363$$
$$(\text{R}^2 = 0.26 \quad \text{sig} < 0.001 \quad \text{N} = 48)$$

(108)

$$\text{Mn} = 3369.383 \times \text{Cd}^2 - 263.720 \times \text{Cd} + 34.107$$
$$(\text{R}^2 = 0.15 \quad \text{sig} < 0.030 \quad \text{N} = 48)$$

(109)

$$\text{Mn} = 20109.848 \times \text{Co}^2 - 652.804 \times \text{Co} + 36.473$$
$$(\text{R}^2 = 0.20 \quad \text{sig} < 0.007 \quad \text{N} = 48)$$

(110)

$$\text{Mn} = 1.807 \times \text{Pb} + 27.597$$
$$(\text{R}^2 = 0.31 \quad \text{sig} < 0.001 \quad \text{N} = 48)$$

(111)

(ix)  Regression equations of copper contribution (Cu)

$$Cu = 20.807 \times N^2 - 18.429 \times N + 3.968$$
$$(R^2 = 0.32 \quad sig < 0.001 \quad N = 48)$$
(112)

$$Cu = 0.00 \times P^2 + 0.030 \times P + 1.858$$
$$(R^2 = 0.18 \quad sig = 0.013 \quad N = 48)$$
(113)

$$Cu = -6.600 \times 10^{-5} \times K^2 + 0.009 \times K + 2.139$$
$$(R^2 = 0.16 \quad sig = 0.019 \quad N = 48)$$
(114)

$$Cu = 3.804 \times 10^{-5} \times Ca^2 - 0.007 \times Ca + 1.870$$
$$(R^2 = 0.11 \quad sig = 0.048 \quad N = 48)$$
(115)

$$Cu = -0.060 \times Zn^2 + 0.611 \times Zn + 1.134$$
$$(R^2 = 0.28 \quad sig < 0.001 \quad N = 48)$$
(116)

$$Cu = 0.001 \times Mn^2 - 0.037 \times Mn + 0.043$$
$$(R^2 = 0.25 \quad sig = 0.003 \quad N = 48)$$
(117)

$$Cu = 1997.327 \times Cr + 5.201$$
$$(R^2 = 0.61 \quad sig < 0.001 \quad N = 48)$$
(118)

$$Cu = 0.145 \times Pb + 1.643$$
$$(R^2 = 0.41 \quad sig = 0.019 \quad N = 48)$$
(119)

$$Cu = 2.541 \times (OM)^2 - 9.554 \times (OM) + 10.844$$
$$(R^2 = 0.67 \quad sig < 0.001 \quad N = 48)$$
(120)

Similarly, the study of the above regression equations from Equations (54)–(120) shows that each group of equations includes the same dependent variable, which is a function of various metals, nutrients, or physical and chemical properties of the soil. Each regression equation within the same group contributes a corresponding amount of the dependent variable according to its interactive orientation (synergistic or antagonistic) and consequently affects the contribution of the dependent variable and, hence, the conditions of soil related to fertility or toxicity.

*3.7. Elemental Interactions and Their Contribution in Heavy Metals to the Experimental Soil of Fescue (Festuca arundinacea Schreb)*

The interactions play an important role in the case of modulating the level of soil toxicity with heavy metals. As it can be seen from the data of Table 8, the regression Equations (121)–(153), mentioned below, contributed to heavy metals recorded in this Table, which clearly show their interactive contribution in terms of heavy metals to the soil.

However, as shown by the contribution of interactions in heavy metals, both their concentrations in soil and the values of the HML (Heavy Metal Load) and EPI (Elemental Pollution Index) pollution indices (Table 8) are generally low, indicating that no toxicity was created in soil due to metal accumulation, owing to the contribution of interactions. These results are due to the low heavy metal content of this experimental soil, so the contribution of these interactions to toxicity was also low. However, in the case of the long-term contribution of heavy metals, even with these small amounts and the continuous reuse of treated wastewater and biosolids, it could easily lead to an increase in metal concentration in the soil, and hence in soil toxicity level, with potential consequences on plant growth and the environment. The interactive accumulation of metals in soils due to natural processes is time-consuming, but it should not be ignored because it can indeed be significant in the long run [6].

3.7.1. Interactive Regression Equations between Heavy metals and Nutrients in the Fescue (*Festuca arundinacea* Schreb) Experimental Soil

(i)　　Regression equations of zinc contribution (Zn)

$$Zn = 13.592 \times Pb^2 - 41.514 \times Pb + 31.635$$
$$(R^2 = 0.72 \quad sig = 0.0013 \quad N = 12) \tag{121}$$

(ii)　　Regression equations of manganese contribution (Mn)

$$Mn = -7.063 \times pH + 61.893$$
$$(R^2 = 0.45 \quad sig = 0.0017 \quad N = 12) \tag{122}$$

$$Mn = 1.315 \times B + 5.131$$
$$(R^2 = 0.68 \quad sig = 0.001 \quad N = 12) \tag{123}$$

(iii)　　Regression equations of copper (Cu)

$$Cu = 2.32 \times C - 5.238$$
$$(R^2 = 0.41 \quad sig = 0.025 \quad N = 12) \tag{124}$$

$$Cu = -0.088 \times (Sludge) + 6.953$$
$$(R^2 = 0.37 \quad sig = 0.036 \quad N = 12) \tag{125}$$

$$Cu = 2.964 \times 10^{-5} \times Mg^2 - 0.024 \times Mg + 7.582$$
$$(R^2 = 0.66 \quad sig = 0.008 \quad N = 12) \tag{126}$$

$$Cu = 0.057 \times Fe + 1.575$$
$$(R^2 = 0.82 \quad sig < 0.001 \quad N = 12) \tag{127}$$

$$Cu = -2.348 \times 10^{-5} \times Na^2 + 0.017 \times Na + 0.810$$
$$(R^2 = 0.64 \quad sig = 0.012 \quad N = 12) \tag{128}$$

(iv)　　Regression equations of cadmium contribution (Cd)

$$Cd = 0.001 \times C + 0.002$$
$$(R^2 = 0.45 \quad sig = 0.018 \quad N = 12) \tag{129}$$

$$Cd = -5.806 \times 10^{-6} \times Si^2 - 0.001 \times Si + 0.062$$
$$(R^2 = 0.54 \quad sig = 0.032 \quad N = 12) \tag{130}$$

$$Cd = -0.046 \times pH + 0.390$$
$$(R^2 = 0.57 \quad sig = 0.004 \quad N = 12) \tag{131}$$

$$Cd = 0.008 \times OM$$
$$(R^2 = 0.57 \quad sig = 0.005 \quad N = 12) \tag{132}$$

$$Cd = 9.536 \times 10^{-8} \times K - 7.938 \times 10^{-5} \times K + 0.035$$
$$(R^2 = 0.67 \quad sig = 0.006 \quad N = 12) \tag{133}$$

$$Cd = 1.446 \times 10^{-7} \times Mg^2 - 8.564 \times 10^{-5} \times Mg + 0.031$$
$$(R^2 = 0.61 \quad sig = 0.013 \quad N = 12) \tag{134}$$

$$Cd = 4.061 \times 10^{-6} \times Ca - 0.008$$
$$(R^2 = 0.46 \quad sig = 0.039 \quad N = 12) \tag{135}$$

$$Cd = 0.001 \times Fe + 0.009$$
$$(R^2 = 0.86 \quad sig = 0.009 \quad N = 12) \tag{136}$$

$$Cd = 0.003 \times Cu^2 - 0.011 \times Cu + 0.028$$
$$(R^2 = 0.83 \quad sig < 0.001 \quad N = 12)$$
(137)

$$Cd = 7.804 \times 10^{-5} \times Na + 0.011$$
$$(R^2 = 0.60 \quad sig = 0.003 \quad N = 12)$$
(138)

$$Cd = -0.016 \times Ni^2 + 0.062 \times Ni - 0.007$$
$$(R^2 = 0.92 \quad sig < 0.001 \quad N = 12)$$
(139)

$$Cd = -3.589 \times 10^{-5} \times Pb^2 + 0.002 \times Pb + 0.019$$
$$(R^2 = 0.77 \quad sig < 0.001 \quad N = 12)$$
(140)

(v)　Regression equations of cobalt contribution (Co)

$$Co = 0.00 \times S^2 - 0.008 \times S + 0.102$$
$$(R^2 = 0.77 \quad sig = 0.004 \quad N = 12)$$
(141)

$$Co = 0.117 \times Ni^2 - 0.133 \times Ni + 0.056$$
$$(R^2 = 0.51 \quad sig = 0.039 \quad N = 12)$$
(142)

(vi)　Regression equations of Nickel contribution (Ni)

$$Ni = 0.015 \times clay + 0.081$$
$$(R^2 = 0.45 \quad sig = 0.016 \quad N = 12)$$
(143)

$$Ni = 0.00 \times (Silt)^2 + 0.009 \times (Silt) + 1.003$$
$$(R^2 = 0.55 \quad sig = 0.027 \quad N = 12)$$
(144)

$$Ni = 2.335 \times 10^{-6} \times Mg^2 - 0.001 \times Mg + 0.499$$
$$(R^2 = 0.63 \quad sig = 0.011 \quad N = 12)$$
(145)

$$Ni = 0.014 \times Fe + 0.245$$
$$(R^2 = 0.89 \quad sig < 0.001 \quad N = 12)$$
(146)

$$Ni = -1.670 \times 10^{-5} \times Na^2 + 0.009 \times Na - 0.11$$
$$(R^2 = 0.69 \quad sig < 0.001 \quad N = 12)$$
(147)

$$Ni = 21.857 \times Cd + 0.068$$
$$(R^2 = 0.92 \quad sig < 0.001 \quad N = 12)$$
(148)

$$Ni = =0.94 \times Pb^2 + 3.419 \times Pb - 2.257$$
$$(R^2 = 0.98 \quad sig < 0.001 \quad N = 12)$$
(149)

(vii)　Regression equations of Lead contribution (Pb)

$$Pb = -0.001 \times Fe^2 + 0.083.Fe + 0.042$$
$$(R^2 = 0.83 \quad sig < 0.001 \quad N = 12)$$
(150)

$$Pb = 0.1198 \times Cu^2 + 1.197 \times Cu - 1.002$$
$$(R^2 = 0.77 \quad sig < 0.001 \quad N = 12)$$
(151)

$$Pb = -3.242 \times 10^{-5} \times Na^2 + 0.017 \times Na - 0.399$$
$$(R^2 = 0.69 \quad sig = 0.005 \quad N = 12)$$
(152)

$$Pb = -1.391 \times Cd2 + 116.143 \times Cd - 0.47$$
$$(R^2 = 0.84 \quad sig < 0.001 \quad N = 12)$$
(153)

**Table 8.** Contribution in heavy metals of interactions between metals, nutrients, and physical and chemical properties of soil under the influence of treated wastewater and under the cultivation of *Festuca arundinacea* Schreb.

| # | Micronutrients and Heavy Metals that Interact | Contribution of Heavy Metal Interactions after the Application of Interventions (mg/kg) | Average Value of Content Soil in Metals (mg/kg) under Control | Difference Contribution of Interactions in Metals (mg/kg) | Percentage Contribution of Interactions (%) |
|---|---|---|---|---|---|
| 1 | Zn | 1.048 | 0.495 | 0.553 | 52.76 |
| 2 | Cu | 3.289 | 2.934 | 0.355 | 10.79 |
| 3 | Mn | 7.800 | 6.35 | 1.45 | 18.59 |
| 4 | Cd | 0.026 | 0.026 | 0.00 | 0.00 |
| 5 | Co | 0.028 | 0.024 | 0.004 | 14.29 |
| 7 | Ni | 0.904 | 0.599 | 0.305 | 33.74 |
| 8 | Pb | 5.22 | 4.15 | 1.07 | 20.50 |
| | Pollution indices | Mean value of pollution indices due to the contribution of Interactions | Value of soil pollution indices under Control | Difference in indices due to the contribution to interactions | Percentage contribution to indices (%) |
| | HML | 1.58 | 1.40 | 0.18 | 11.39 |
| | EPI | 0.81 | 0.31 | 0.52 | 64.20 |

The values of the pollution indices due to the interactive contribution in terms of heavy metals that took place in the soil of the *Festuca arundinacea* Schreb of Messolonggi experiment, as described in Table 8, are low to very low, not reflecting any soil toxicity. Similar results were found by the calculated low pollution indices, which were verified by the experimental data of lettuce (*Lactuca sativa* var. *Longifolia*) soil [8] reported in Table 9. These low attained toxicity values suggested by the pollution indices, differed slightly from those of the Fescue experiment due only to the variability of soil microclimatic conditions between the two experiments, as well as due to the different experimental plants studied. However, the general trend of the interactive effects on metals in both soils, nutrients, and soil properties, were more or less similar in contributing heavy metals and plant nutrients changing the soil fertility and toxicity approximately similarly at about the same level in both cases of Fescue and Lettuce experiments. However, Zn, Mn, and Cu were being contributed at higher levels, compared to other nutrients, and heavy metals, in both of the above cases (Tables 8 and 9).

As far as the values of the pollution indices are concerned, the results obtained under the effect of interactions in the soil of the Fescue experiment for the indices HML and EPI were 1.58. and 0.81, respectively (Table 8), while, in the lettuce experiment, the values of HML and EPI were 1.71 and 1.06, respectively (Table 9). These values reflected low to light soil toxicity, which generally caused minor losses of plant yields [6].

**Table 9.** Contribution in heavy metals by the interactions between metals, nutrient, physical and chemical properties of soil under the influence of treated wastewater and biosolid in the presence of cultivation of Lettuce (*Lactuca sativa* var. *Longifolia*).

| # | Interacting Heavy Metals | Contribution in Metals by the Heavy Metal Interactions (mg/kg) | Mean Level of Soil Heavy Metals (mg/kg) | The Actual Contribution by the Heavy Metal Interactions (mg/kg) | Percent (%) Contribution in Heavy Metals (mg/kg) |
|---|---|---|---|---|---|
| 1 | Zn | 2.29 | 2.24 | 0.05 | 2.18 |
| 2 | Cu | 2.50 | 2.12 | 0.38 | 15.20 |
| 3 | Mn | 33.09 | 33.50 | −0.41 | −1.24 |
| 4 | Cr | 0.054 | 0.050 | 0.004 | 7.41 |
| 5 | Co | 0.096 | 0.050 | 0.046 | 47.20 |
| 6 | Ni | 0.361 | 0.132 | 0.229 | 63.43 |
| 7 | Pb | 4.19 | 3.27 | 0.92 | 21.96 |
| | Pollution Index | Mean contribution to the indices due to interactions | Value of indices under Control | Differences due to interactions contribution | Percent contribution to indices (%) |
| | HML | 1.71 | 1.70 | 0.01 | 0.58 |
| | EPI | 1.06 | 0.74 | 0.32 | 30.19 |

The study of Table 9 reveals that the percent contribution of interactions in some heavy metals was high. Nevertheless, based on the pollution indices values HML and EPI, the relatively high percent contribution in Ni 63.43%, Co 47.20%, and Pb 21.96% did not cause any substantial increase in soil toxicity since as mentioned above, the original levels of these metals in the soil were very low (Ni 0.361, Co 0.096 and Pb 4.19 mg/kg (Table 9) as in the case of fescue experiment soil (Cd 0.026, Co 0.024, Pb 4.15 and Ni 0.599 mg/kg).

### 3.7.2. Regression Equations between Heavy Metals and Plant Nutrients and Their Contribution in Heavy Metals to the Experimental Soil of Lettuce (*Lactuca sativa* var. *Longifolia*) (1st Soil Sampling)

The regression equations mentioned below from No (154) to (207), based on their high statistical significance, have been arranged in groups of varying numbers of equations, where each group has the same dependent variable. This arrangement gives the possibility to solve each equation separately and to calculate the mean contribution in each heavy element, helping us to calculate the mean contribution of each group of equations in metal, thus, giving a holistic view about the contributed element, respectively, for each group, as a function of various elements and soil properties.

These interactive activities may affect soil fertility and soil toxicity. The careful study of these interactions shows that the various heavy metals may interact with other metals, macro, and micronutrients, impacting soil fertility and toxicity, respectively, depending on their interactive capacity. A meticulous examination of the below-stated regression equations reveals that the heavy metals interact variably with various inter-acting factors such as heavy metals, nutrients, and soil physical and chemical properties. For example, Zn interacted with Mn, pH, P, K, and Ca. Also, Mn with pH, P, K, Ca, Mg, Fe, Zn, Cu, Cr, Cd, Co, Pb, etc. On the other hand, Co interacted only with Mn, Ni, and pH. These variable interactive differences indicate that the elements interact with various factors (be they metals or nutrients), depending on the chemical affinity, the concentration of the interacting elements, the level of pH, OM, Clay, oxides of Fe and Mn, absorption/desorption, and redox potential [19]. All these factors affect the behavior of the elements in the soil, acting as a regulating agent by means of their interactive activity.

### 3.7.3. Regression Equations Contributing Exclusively Interactive Heavy Metals in the Soil of the *Lactuca sativa* Lettuce Experiment

(i)    Regression equations of Zinc contribution (Zn)

$$Zn = 0.001 \times EC^2 - 0.051 \times EC + 2.173$$
$$(R^2 = 0.43 \quad sig < 0.001 \quad N = 48)$$
(154)

$$Zn = -0.001 \times Mg^2 + 0.100 \times Mg + 0.671$$
$$(R^2 = 0.13 \quad sig = 0.041 \quad N = 48)$$
(155)

$$Zn = 0.006 \times Fe^2 - 0.065 \times Fe + 0.579$$
$$(R^2 = 0.22 \quad sig = 0.003 \quad N = 48)$$
(156)

$$Zn = 0.063 \times Mn + 0.132$$
$$(R^2 = 0.14 \quad sig = 0.010 \quad N = 48)$$
(157)

$$Zn = -1.199 \times Cu^2 + 7.191 \times Cu - 7.305$$
$$(R^2 = 0.23 \quad sig = 0.003 \quad N = 48)$$
(158)

$$Zn = 50.753 \times Cr - 0.293$$
$$(R^2 = 0.10 \quad sig = 0.030 \quad N = 48)$$
(159)

$$Zn = 184.899 \times Ni^2 - 93.098 \times Ni + 13.460$$
$$(R^2 = 0.23 \quad sig = 0.003 \quad N = 48)$$
(160)

(ii) Regression equations of Manganese contribution (Mn)

$$Mn = -1.195 \times pH^2 + 9.972 \times pH + 16.910$$
$$(R^2 = 0.23 \quad sig = 0.003 \quad N = 48)$$

(161)

$$Mn = -0.003 \times p^2 + 0.787 \times P + 0.526$$
$$(R^2 = 0.46 \quad sig < 0.001 \quad N = 48)$$

(162)

$$Mn = -0.002 \times K^2 + 0.535 \times K + 3.223$$
$$(R^2 = 0.47 \quad sig < 0.001 \quad N = 48)$$

(163)

$$Mn = -0.001 \times Ca^2 + 0.314 \times Ca + 0.836$$
$$(R^2 = 0.45 \quad sig < 0.001 \quad N = 48)$$

(164)

$$Mn = \ln(Mg) \times 5.106 + 17.264$$
$$(R^2 = 0.56 \quad sig < 0.001 \quad N = 48)$$

(165)

$$Mn = -0.034 \times Fe^2 + 2.264 \times Fe + 0.362$$
$$(R^2 = 0.52 \quad sig < 0.001 \quad N = 48)$$

(166)

$$Mn = \ln(Zn) \times 7.321 + 28.679$$
$$(R^2 = 0.48 \quad sig < 0.001 \quad N = 48)$$

(167)

$$Mn = 3.735 \times Cu^2 - 13.842 \times Cu + 45.599$$
$$(R^2 = 0.14 \quad sig = 0.031 \quad N = 48)$$

(168)

$$Mn = 20273.705 \times Cr^2 - 1714.287 \times Cr + 67.363$$
$$(R^2 = 0.26 \quad sig = 0.001 \quad N = 48)$$

(169)

$$Mn = 3369.383 \times Cd^2 - 263.720 \times Cd + 34.107$$
$$(R^2 = 0.14 \quad sig = 0.030 \quad N = 48)$$

(170)

$$Mn = 20109.848 \times Co^2 - 652.804 \times Co + 36.473$$
$$(R^2 = 0.20 \quad sig = 0.007 \quad N = 48)$$

(171)

$$Mn = 1.807 \times Pb + 27.597$$
$$(R^2 = 0.31 \quad sig < 0.001 \quad N = 48)$$

(172)

(iii) Regression equations of Copper contribution (Cu)

$$Cu = 20.807 \times N^2 - 18.429 \times N + 3.968$$
$$(R^2 = 0.32 \quad sig < 0.001 \quad N = 48)$$

(173)

$$Cu = 0.00 \times P^2 + 0.030 \times P + 1.858$$
$$(R^2 = 0.18 \quad sig = 0.013 \quad N = 48)$$

(174)

$$Cu = -6.600 \times 10^{-5} \times K^2 + 0.009 \times K + 2.139$$
$$(R^2 = 0.16 \quad sig = 0.019 \quad N = 48)$$

(175)

$$Cu = 3.804 \times 10^{-5} \times Ca^2 - 0.007 \times Ca + 1.870$$
$$(R^2 = 0.11 \quad sig = 0.048 \quad N = 48)$$

(176)

$$Cu = -0.060 \times Zn^2 + 0.611 \times Zn + 1.134$$
$$(R^2 = 0.28 \quad sig < 0.001 \quad N = 48)$$

(177)

$$Cu = 0.001 \times Mn^2 - 0.037 \times Mn + 0.043$$
$$(R^2 = 0.23 \quad sig = 0.003 \quad N = 48)$$

(178)

$$Cu = 0.145 \times Pb + 1.643$$
$$(R^2 = 0.41 \quad sig = 0.019 \quad N = 48)$$
(179)

$$Cu = 2.541 \times (OM)^2 - 9.554 \times (OM) + 10.844$$
$$(R^2 = 0.67 \quad sig < 0.001 \quad N = 48)$$
(180)

(iv)    Regression equations of Chromium contribution (Cr)

$$Cr = 0.222 \times N^2 - 0.199 \times N + 0.070$$
$$(R^2 = 0.15 \quad sig = 0.024 \quad N = 48)$$
(181)

$$Cr = -5.416 \times 10^{-6} \times P^2 + 0.001 \times P + 0.042$$
$$(R^2 = 0.31 \quad sig < 0.001 \quad N = 48)$$
(182)

$$Cr = -0.001 \times Zn^2 + 0.008 \times Zn + 0.037$$
$$(R^2 = 0.21 \quad sig = 0.004 \quad N = 48)$$
(183)

$$Cr = 0.00001189 \times Mn^2 + 0.00 \times Mn + 0.043$$
$$(R^2 = 0.23 \quad sig = 0.003 \quad N = 48)$$
(184)

$$Cr = -0.002 \times Cu^2 - 0.001 \times Cu + 0.041$$
$$(R^2 = 0.47 \quad sig < 0.001 \quad N = 48)$$
(185)

$$Cr = 0.536 \times Ni^2 - 0.348 \times Ni + 0.014$$
$$(R^2 = 0.17 \quad sig = 0.016 \quad N = 48)$$
(186)

$$Cr = 0.001 \times Pb^2 - 0.011 \times Pb + 0.027$$
$$(R^2 = 0.44 \quad sig < 0.001 \quad N = 48)$$
(187)

$$Cr = 0.023 \times (OM)^2 - 0.091 \times (OM) + 0.135$$
$$(R^2 = 0.36 \quad sig < 0.001 \quad N = 48)$$
(188)

$$Cr = 0.536 \times Ni^2 - 0.348 \times Ni + 0.104$$
$$(R^2 = 0.17 \quad sig = 0.016 \quad N = 48)$$
(189)

$$Cr = -0.001 \times Pb^2 + 0.0110 \times Pb + 0.027$$
$$(R^2 = 0.44 \quad sig < 0.001 \quad N = 48)$$
(190)

(v)    Regression equations of cobalt contribution (Co)

$$Co = -0.00001366 \times Mn^2 + 0.00 \times Mn + 0.016$$
$$(R^2 = 0.23 \quad sig = 0.003 \quad N = 48)$$
(191)

$$Co = 0.065 \times Ni + 0.005$$
$$(R^2 = 0.12 \quad sig = 0.008 \quad N = 48)$$
(192)

$$Co = -0.016 \times pH^2 + 0.092 \times pH + 0.277$$
$$(R^2 = 0.16 \quad sig = 0.022 \quad N = 48)$$
(193)

(vi)    Regression equations of Nickel contribution (Ni)

$$Ni = -1.367 \times N^2 + 1.216 \times N + 0.158$$
$$(R^2 = 0.18 \quad sig = 0.011 \quad N = 48)$$
(194)

$$Ni = 0.00001819 \times P^2 - 0.002 \times P + 0.252$$
$$(R^2 = 0.19 \quad sig = 0.010 \quad N = 48)$$
(195)

$$Ni = 0.001 \times Fe^2 - 0.011 \times Fe + 0.245$$
$$(R^2 = 0.64 \quad sig < 0.001 \quad N = 48)$$
(196)

$$Ni = -0.015 \times Cu^2 + 0.043 \times Cu + 0.261$$
$$(R^2 = 0.17 \quad sig = 0.016 \quad N = 48)$$
(197)

$$Ni = -39.106 \times Cr^2 + 3.071 \times Cr + 0.277$$
$$(R^2 = 0.13 \quad sig = 0.040 \quad N = 48)$$
(198)

$$Ni = 0.941 \times Cd + 0.212$$
$$(R^2 = 0.10 \quad sig = 0.031 \quad N = 48)$$
(199)

$$Ni = 73.731 \times Co^2 - 1.926 \times Co + 0.281$$
$$(R^2 = 0.13 \quad sig < 0.048 \quad N = 48)$$
(200)

$$Ni = -0.109 \times (OM)^2 + 0.436 \times (OM) + 0.090$$
$$(R^2 = 0.29 \quad sig < 0.001 \quad N = 48)$$
(201)

(vii)  Regression equations of lead contribution (Pb)

$$Pb = 60.816 \times N^2 - 51.923 \times N + 8.421$$
$$(R^2 = 0.16 \quad sig < 0.019 \quad N = 48)$$
(202)

$$Pb = 0.00 \times K^2 + 0.408 \times K + 2.815$$
$$(R^2 = 0.17 \quad sig < 0.015 \quad N = 48)$$
(203)

$$Pb = -0.001 \times Mg^2 + 0.140 \times Mg + 0.399$$
$$(R^2 = 0.16 \quad sig < 0.019 \quad N = 48)$$
(204)

$$Pb = -0.324 \times Zn^2 + 2.928 \times Zn - 1.184$$
$$(R^2 = 0.26 \quad sig < 0.001 \quad N = 48)$$
(205)

$$Pb = 0.006 \times Mn^2 - 0.176 \times Pb + 1.761$$
$$(R^2 = 0.48 \quad sig < 0.001 \quad N = 48)$$
(206)

$$Pb = 0.260 \times Cu^2 + 4.175 \times Cu - 4.341$$
$$(R^2 = 0.41 \quad sig < 0.001 \quad N = 48)$$
(207)

3.7.4. Interactions between Metals and Chemical and Physical Soil Properties and Their Contribution to the Characteristics of *Festuca arundinacea* Schreb Experimental Soil

As the elemental interactions affect not only the various processes and phenomena of soil, they also exert an important and significant impact on its physical and chemical properties [4]. The following section deals with this critical aspect of the elemental interactive activities.

The regression analysis between soil physical and chemical properties, metals, and nutrients produced several statistically significant regression equations. Thus, it was found that sand (S), silt (Si), clay (C), EC, OM, and pH of the soil were significantly affected by their interactions with metals and nutrients. The following statistically significant regression equations from Equations (208)–(230) clearly describe these interactions and their relationships with soil's physical and chemical factors. The study of the data of Table 10, mentioned below, based on the mean contribution of interactions, shows that, in general, the above physical and chemical abiotic factors interacted both antagonistically (Clay, pH) and synergistically (sand, Si, OM, and $CaCO_3$) with soil fine sand, silt, heavy metals, and micronutrients. As shown by the below-listed regression equations, the Clay and pH mean contribution values were negative due to the interference of antagonistic interactions between "Fine sand x Clay" (Figure 1) and "Silt x Clay" (Figure 2).

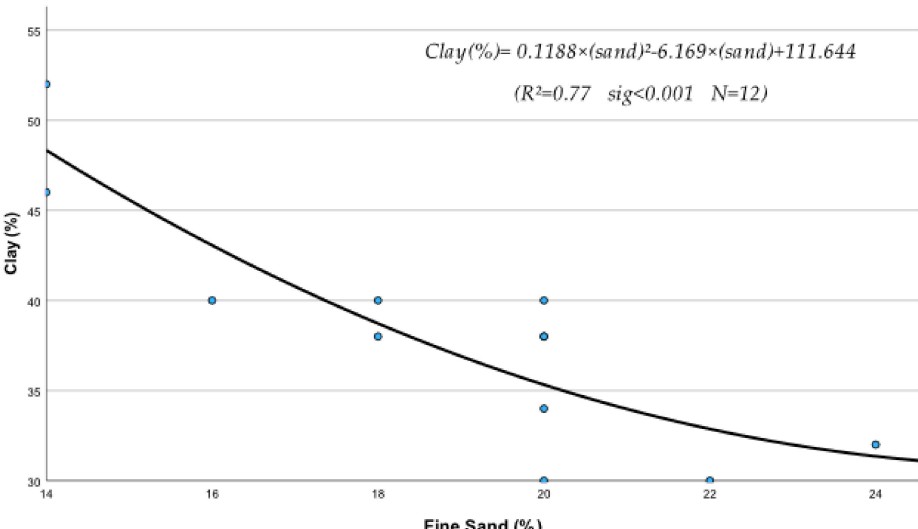

**Figure 1.** Interaction between fine sand and clay particles of soil under the effect of treated wastewater and biosolids in the presence of growing fescue plants.

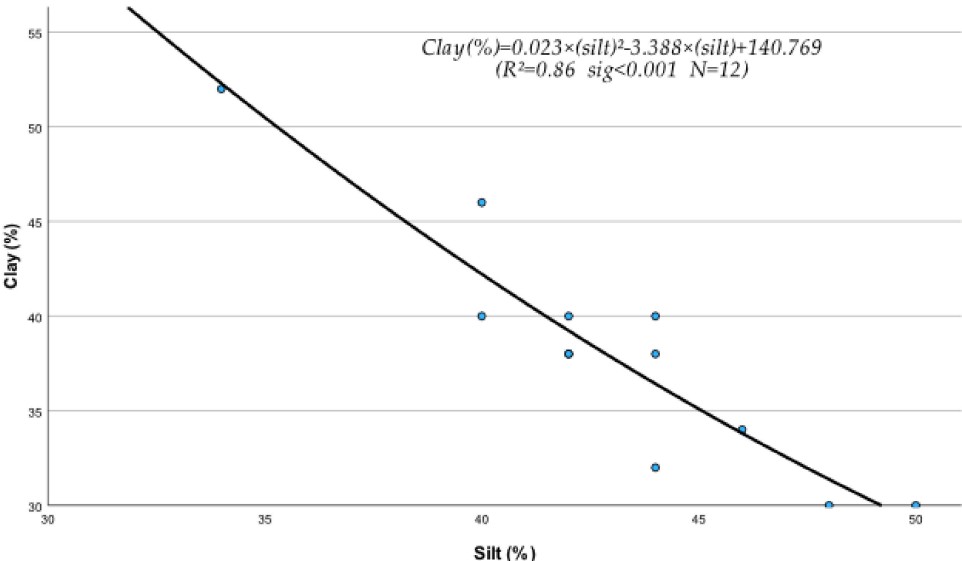

**Figure 2.** Interaction between fine sand and silt particles of soil under the effect of treated wastewater and biosolids in the growing Fescue plants.

Also, the interactions of "Clay x Sludge" (Figure 2) and "pH x Clay" (Figure 3) were antagonistic, which contributed to the generation of antagonistic results. Note that, in general, such antagonistic interactions are the cause of the mean negative effects of the interactively produced results in many cases.

These outcomes show beyond doubt that in soil, everything is subject to interactions, which may supply small or large amounts of metals and nutrients, organic matter, calcium carbonate, etc., and increase or decrease the soil properties (Table 10). Studying the data of this Table shows that the interactions contributed the following percent changes to the soil properties:

1. Increase of very fineS by +27.83%.
2. Si by +0.12%,
3. OM by +0.49%, and calcium carbonate by +0.41%.

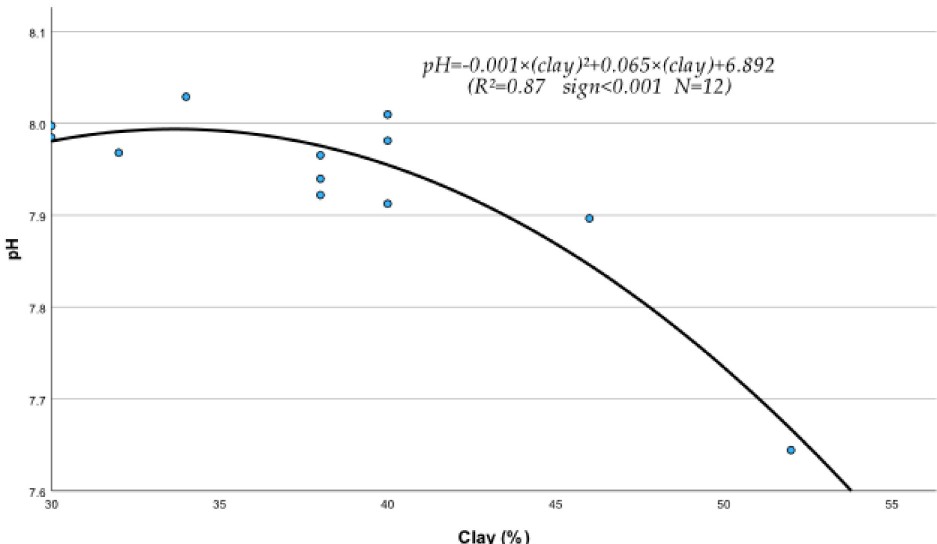

$$pH=-0.001\times(clay)^2+0.065\times(clay)+6.892$$
$$(R^2=0.87 \quad sign<0.001 \quad N=12)$$

**Figure 3.** Interaction between clay particles and pH of soil under the effect of treated wastewater and biosolids in the presence of growing Fescue plants.

On the other hand, the antagonistic interactions decreased C by $-77.45\%$ and pH by $-1.41\%$ (Table 10).

**Table 10.** Changes in soil's physical and chemical properties because of interactions between metals and nutrients, under the influence of treated waste water and biosolids, and under the cultivation of *Festuca arundinacea* Schreb.

| # | Soil Properties | Effect of Interactions on the Mean Soil Properties Values | Mean Value of Soil Properties | Contribution of Interactions to Soil Properties | Percent Change of Soil Properties |
|---|---|---|---|---|---|
| 1 | Sand (S) (%) | 46.30 | 18.47 | +27.83 | +60.11 |
| 2 | Silt (Si) (%) | 43.46 | 43.00 | +0.12 | +0.28 |
| 3 | Clay (C) (%) | 21.51 | $-38.17$ | $-16.66$ | $-77.45$ |
| 4 | OM (%) | 3.75 | 3.26 | +0.49 | +13.07 |
| 6 | CaCO$_3$ (%) | 5.54 | 5.13 | +0.41 | +7.40 |
| 7 | pH$_{(sat\ paste)}$ | 7.83 | 7.94 | $-0.11$ | $-1.41$ |

As shown in Table 10, the significant regression Equations (222)–(224), expressed as a function of Fe, Cd, and Pb, respectively, contributed to the OM a mean quantity equal to 0.41%. This increase corresponds to 13.07% of the mean percent inter-active value (see Section 3.7.4 (iv)).

It must be mentioned, in this respect, that though a metal may interact with several elements, the outcome of these interactions may be negative if, for example, one or more interactions are antagonistic. In fact, in the case of Clay, out of its six interactions, the two, i.e., Clay x Sand and Clay x Silt as mentioned before, were found to be highly antagonistic (Figures 1 and 2) and as a result, the "percent interactive contribution" in terms of Clay was also negative. These results suggest that the interaction of Clay with silt and fine sand contributes to the removal and decrease in these soil textural components A recently published paper referring to a study of internal erosion test, reported amongst others that the Clay of a mixture of "clay-sand-gravel" caused enormous particle loss, suggesting that it may decrease sand particle of the mixture [37]. Similarly, the interaction between pH and Clay was also antagonistic (negative), leading to a decrease in Clay (Figure 3). The latter may be due to the effect of pH on the mineralogical composition of Clay, which may cause, with time, break up and disintegration of the soil clay mineral composition.

The presence of antagonistic interactions responsible for the outcome of the antagonistic interactive contribution may explain the negative effect of the Clay. If, however, the percent antagonism is low, then the final result could be positive, meaning that the synergistic interactions contributed higher levels than the antagonist ones.

3.7.5. Regression Equations between Metals and Chemical and Physical Soil Properties Related to the *Festuca arundinacea* Schreb Experimental Soil

(i)     Regression equations of fine sand contribution (S)

$$S = -0.402 \times (C) + 34.185$$
$$(R^2 = 0.73 \quad \text{sig} < 0.001 \quad N = 12) \tag{208}$$

$$S = -58015.980 \times Cd^2 + 2790.215 \times Cd -12.53$$
$$(R^2 = 0.52 \quad \text{sig} = 0.036 \quad N = 12) \tag{209}$$

$$S = -11.720 \times \ln(Co) - 27.125$$
$$(R^2 = 0.42 \quad \text{sig} = 0.023 \quad N = 12) \tag{210}$$

(ii)     Regression equations of sludge contribution (Si)

$$Si = 0.001 \times (C)^2 - 0.657 \times (C) + 66.975$$
$$(R^2 = 0.86 \quad \text{sig} < 0.001 \quad N = 12) \tag{211}$$

$$Si = 0.823 \times S + 27.505$$
$$(R^2 = 0.36 \quad \text{sig} = 0.040 \quad N = 12) \tag{212}$$

$$Si = -0.008 \times Fe^2 + 0.137 \times Fe + 46.439$$
$$(R^2 = 0.67 \quad \text{sig} = 0.007 \quad N = 12) \tag{213}$$

$$Si = -6.746 \times Cu^2 + 38.860 \times Cu - 10.150$$
$$(R^2 = 0.58 \quad \text{sig} = 0.023 \quad N = 12) \tag{214}$$

$$Si = -45788.150 \times Cd^2 + 1867.883 \times Cd + 26.937$$
$$(R^2 = 0.65 \quad \text{sig} = 0.009 \quad N = 12) \tag{215}$$

(iii)   Regression equations of clay contribution (C)

$$C = 0.118 \times S^2 - 6.169 \times S + 111.644$$
$$(R^2 = 0.78 \quad \text{sig} < 0.001 \quad N = 12) \tag{216}$$

$$C = 0.023 \times Si^2 - 3.388 \times Si + 140.769$$
$$(R^2 = 0.87 \quad \text{sig} < 0.001 \quad N = 12) \tag{217}$$

$$C = 0.018 \times Fe^2 - 0.536 \times Fe + 37.766$$
$$(R^2 = 0.60 \quad \text{sig} = 0.016 \quad N = 12) \tag{218}$$

$$C = 9.180 \times Cu^2 - 51.756 \times Cu + 106.816$$
$$(R^2 = 0.57 \quad \text{sig} = 0.023 \quad N = 12) \tag{219}$$

$$C = 103804.13 \times Cd^2 + 4658.098 \times Cd + 85.594$$
$$(R^2 = 0.69 \quad \text{sig} = 0.005 \quad N = 12) \tag{220}$$

$$C = 23.241 \times \ln(Co) +129.307$$
$$(R^2 = 0.36 \quad \text{sig} = 0.0382 \quad N = 12) \tag{221}$$

(iv)   Equations of covariation of organic substance (OM)

$$OM = 0.038 \times Fe + 2.184$$
$$(R^2 = 0.38 \qquad sig = 0.033 \qquad N = 12)$$

(222)

$$OM = 1101.073 \times Cd^2 + 15.409 \times Cd + 2.078$$
$$(R^2 = 0.57 \qquad sig = 0.022 \qquad N = 12)$$

(223)

$$OM = 1.837 \times \ln(Pb) + 2.473$$
$$(R^2 = 0.37 \qquad sig = 0.035 \qquad N = 12)$$

(224)

(v)    Regression equations of pH contribution ($pH_{sat\ paste}$)

$$pH = -0.001 \times C^2 + 0.082 \times C + 6.217$$
$$(R^2 = 0.66 \qquad sig = 0.008 \qquad N = 12)$$

(225)

$$pH = -0.002 \times Si^2 + 0.212 \times Si + 3.337$$
$$(R^2 = 0.86 \qquad sig < 0.001 \qquad N = 12)$$

(226)

$$pH = -0.001 \times Fe^2 + 0.035 \times Fe + 7.332$$
$$(R^2 = 0.69 \qquad sig = 0.005 \qquad N = 12)$$

(227)

$$pH = -0.036 \times Mn^2 + 0.434Mn + 6.501$$
$$(R^2 = 0.53 \qquad sig = 0.075 \qquad N = 12)$$

(228)

$$pH = -0.165 \times B^2 + 0.330 \times B + 7.667$$
$$(R^2 = 0.59 \qquad sig = 0.019 \qquad N = 12)$$

(229)

(vi)   Regression equations of calcium carbonate contribution ($CaCO_3$)

$$CaCO_3 = -0.334 \times NO_3 + 15.897$$
$$(R^2 = 0.40 \qquad sig = 0.029 \qquad N = 12)$$

(230)

3.7.6. Interactions between Metals, and Physical and Chemical Properties of Soil, and Their Contribution to the Characteristics of the Lettuce (*Lactuca sativa* var. *Longifolia*) Experimental Soil

It was considered necessary for reasons of comparison, to study the contribution of the interactions between "physical, chemical soil properties, heavy metals, and, nutrients", in the experimental soil of *Festuca arundinacea* Schreb. whose relevant analytical data were mentioned above, and also the data of the lettuce experimental soil, [35] the experiment having been conducted in Agrinion, University of Ioannina, Greece, under our supervision. From the statistical processing of these data by means of regression analysis of the soil analytical data at its 1st and 2nd soil sampling, the following regression equations were obtained. i.e., from Equations (231)–(241) at the 1st and from Equations (242)–(261) at the 2nd soil sampling, respectively, where the various characteristic properties of the soil are given as a function of the macro and micronutrients.

The study of the data reported in Table 10 reveals a typical picture of the elemental interactions' contribution to the properties of soil. The contributions attained are both high and low or zero, a fact, which already has been observed in the previous cases of contribution to soil characteristics of *Festuca arundinacea* Schreb. Overall, these results reflect the dynamic nature of the interactions.

Table 11 presents the contribution of interactions to the chemical and physical properties of the soil of the lettuce experiment at its two-soil samplings. It was found that the EC increased during the first sampling by 77.70%, while during the second soil sampling, the EC decreased to 73.33%. In other words, the EC decreased by −3.33%. On the other hand, the pH decreased by −0.18% during both soil samplings (Table 11) and on the contrary, the organic matter increased by the contribution of interactions by 0.60% during both the first and second sampling, respectively, corresponding to 465kg OM/ha/0–30cm soil depth,

equivalent to 11.6kg of total N/ha/year. This increase was due to the interactions of OM with N, Fe, Cr, Ni, and P for the soil of the first sampling, while the OM interacted with pH, N, K, Mn, Cu, Cr, Ni, and Pb in the second sampling, producing the same quantity of OM and total N/ha as in the first sampling.

**Table 11.** Changes in the physical and chemical properties of soil as a result of interactions between metals and nutrients, under the influence of treated wastewater and biosolids and in the presence of lettuce culture (*Lactuca sativa* var. *Longifolia*) at the 1st and 2nd soil sampling.

| Physical and Chemical Soil Properties | Effect of Interactions on the Mean Level of Soil Properties | Mean Soil Properties Level | Actual Contribution of Interaction to Soil Properties | Percent Changes of Soil Properties Level |
|---|---|---|---|---|
| 1st Soil sampling | | | | |
| pH | 5.61 | 5.62 | −0.01 | −0.18 |
| EC (S/cm) | 4.08 | 0.91 | 3.17 | +77.70 |
| OM (%) | 1.66 | 1.65 | 0.01 | +0.60 |
| 2nd Soil sampling | | | | |
| pH | 5.61 | 5.62 | −0.01 | −0.18 |
| EC (S/cm) | 3.00 | 0.80 | 2.20 | +73.33 |
| OM (%) | 1.66 | 1.65 | 0.01 | +0.60 |

3.7.7. Regression Equations between Soil Properties with Metals, Nutrients and Soil Characteristics during 1st and 2nd Sampling of the Lettuce (*Lactuca sativa* var. *Longifolia*) Experimental Soil

Regression Equations of the 1st Soil Sampling

(i)   Regression equations of pH contribution (pH)

$$pH = \ln(Mg) \times 0.655 + 3.553$$
$$(R^2 = 0.33 \quad \text{sig} < 0.001 \quad N = 48) \tag{231}$$

$$pH = -0.010 \times Fe^2 + 0.481 \times Fe + 0.186$$
$$(R^2 = 0.46 \quad \text{sig} < 0.001 \quad N = 48) \tag{232}$$

$$pH = \ln(Zn) \times 1.015 + 4.968$$
$$(R^2 = 0.34 \quad \text{sig} < 0.001 \quad N = 48) \tag{233}$$

$$pH = -0.004 \times Mn^2 + 0.289 \times Mn + 0.537$$
$$(R^2 = 0.42 \quad \text{sig} < 0.001 \quad N = 48) \tag{234}$$

(ii)   Regression equations of Electrical conductivity contribution (EC)

$$EC = 2.173 \times Zn^2 - 10.509 \times Zn + 12.243$$
$$(R^2 = 0.70 \quad \text{sig} < 0.001 \quad N = 48) \tag{235}$$

(iii)   Regression equations of Organic matter contribution (OM)

$$OM = -6.472 \times N^2 + 5.241 \times N + 1.141$$
$$(R^2 = 0.25 \quad \text{sig} = 0.002 \quad N = 48) \tag{236}$$

$$OM = 0.002 \times Fe^2 - 0.051 \times Fe + 1.823$$
$$(R^2 = 0.17 \quad \text{sig} = 0.014 \quad N = 48) \tag{237}$$

$$OM = 0.003 \times Cu^2 + 0.347 \times Cu + 2.374$$
$$(R^2 = 0.57 \quad \text{sig} < 0.001 \quad N = 48) \tag{238}$$

$$OM = -598.972 \times Cr^2 + 46.396 \times Cr + 0.865$$
$$(R^2 = 0.39 \quad \text{sig} = 0.001 \quad N = 48) \tag{239}$$

$$\text{OM} = -14.633 \times \text{Ni}^2 + 10.407 \times \text{Ni} - 0.088$$
$$(R^2 = 0.30 \qquad \text{sig} < 0.001 \qquad N = 48)$$
(240)

$$\text{OM} = 0.006 \times \text{Pb}^2 - 0.114 \times \text{Pb} + 1.933$$
$$(R^2 = 0.32 \qquad \text{sig} < 0.001 \qquad N = 48)$$
(241)

Regression Equations of the 2nd Soil Sampling

(i)     Regression equations of pH contribution(pH)

$$\text{pH} = 0.798 \times \text{N} + 5.517$$
$$(R^2 = 0.99 \qquad \text{sig} < 0.001 \qquad N = 48)$$
(242)

$$\text{pH} = 0.010 \times \text{Fe}^2 - 0.130 \times \text{Fe} + 3.131$$
$$(R^2 = 0.99 \qquad \text{sig} < 0.001 \qquad N = 48)$$
(243)

$$\text{pH} = 0.011 \times \text{Zn}^2 + 0.137 \times \text{Zn} + 5.248$$
$$(R^2 = 0.99 \qquad \text{sig} < 0.001 \qquad N = 48)$$
(244)

$$\text{pH} = 0.014 \times \text{Cu}^2 + 0.110 \times \text{Cu} + 5.326$$
$$(R^2 = 0.86 \qquad \text{sig} < 0.001 \qquad N = 48)$$
(245)

$$\text{pH} = 0.828 \times \text{Cr} + 5.582$$
$$(R^2 = 0.99 \qquad \text{sig} < 0.001 \qquad N = 48)$$
(246)

(ii)     Regression equations of electrical conductivity contribution (EC)

$$\text{EC} = 1.114 \times \text{pH} - 3.330$$
$$(R^2 = 0.60 \qquad \text{sig} < 0.001 \qquad N = 48)$$
(247)

$$\text{EC} = 0.891 \times \text{N} + 2.808$$
$$(R^2 = 0.60 \qquad \text{sig} = 0.001 \qquad N = 48)$$
(248)

$$\text{EC} = 0.009 \times \text{Fe}^2 - 0.147 \times \text{Fe} - 5.344$$
$$(R^2 = 0.62 \qquad \text{sig} < 0.001 \qquad N = 48)$$
(249)

$$\text{EC} = 0.025 \times \text{Mn}^2 - 1.27 \times \text{Mn} + 16.525$$
$$(R^2 = 0.54 \qquad \text{sig} < 0.001 \qquad N = 48)$$
(250)

$$\text{EC} = 0.002 \times \text{Cu}^2 + 1.197\text{Cu} + 0.406$$
$$(R^2 = 0.60 \qquad \text{sig} < 0.001 \qquad N = 48)$$
(251)

$$\text{EC} = 0.924 \times \text{Cr} + 2.880$$
$$(R^2 = 0.60 \qquad \text{sig} < 0.001 \qquad N = 48)$$
(252)

(iii)     Regression equations of Organic matter contribution (OM)

$$\text{OM} = -0.006 \times \text{pH} + 1.697$$
$$(R^2 = 0.12 \qquad \text{sig} = 0.016 \qquad N = 48)$$
(253)

$$\text{OM} = -0.005 \times \text{N} + 1.665$$
$$(R^2 = 0.12 \qquad \text{sig} = 0.017 \qquad N = 48)$$
(254)

$$\text{OM} = 0.00003076 \times \text{K}^2 - 0.005 \times \text{K} + 1.747$$
$$(R^2 = 0.13 \qquad \text{sig} = 0.043 \qquad N = 48)$$
(255)

$$\text{OM} = -0.00008992 \times \text{Mg}^2 + 0.004 \times \text{Mg} + 1.655$$
$$(R^2 = 0.17 \qquad \text{sig} = 0.016 \qquad N = 48)$$
(256)

$$OM = -0.010 \times Mn + 1.985$$
$$(R^2 = 0.21 \quad sig = 0.001 \quad N = 48)$$

(257)

$$OM = 0.004 \times Cu^2 - 0.336 \times Cu + 2.349$$
$$(R^2 = 0.59 \quad sig < 0.001 \quad N = 48)$$

(258)

$$OM = -0.005 \times Cr + 1.664$$
$$(R^2 = 0.12 \quad sig = 0.017 \quad N = 48)$$

(259)

$$OM = -14.633 \times Pb^2 + 10.407 \times Ni - 0.088$$
$$(R^2 = 0.30 \quad sig < 0.001 \quad N = 48)$$

(260)

$$OM = 0.006 \times Pb^2 - 0.114 \times Pb + 1.933$$
$$(R^2 = 0.32 \quad sig = 0.001 \quad N = 48)$$

(261)

## 4. Conclusions

The conclusions drawn from the above-mentioned are as follows:

1. Hundreds of elemental interactions between metals, macro micronutrients, and the physical and chemical properties of soil occur in the soil environment, which contribute to heavy metals and nutrients affecting soil fertility and toxicity.
2. Depending on their interactive synergistic or antagonistic orientation of the regression equations, they contribute quantitatively and affect positively or negatively their interacting dependent variable, hence influencing the function of soils.
3. In the present work, the elemental interactions not only contributed to fescue and lettuce soil considerable quantities of plant nutrients. It was also possible to quantitatively evaluate them, expressed in the form of corresponding fertilizers accumulating in the soil, i.e., in the fescue experiment, 7.80 kg/ha of potassium sulfate (0-0-50) and 19.6 kg/ha of borax, while in the lettuce soil the fertilizers accumulated were: 57.6 kg/ha of supper phosphate (0-46-0); and 31.3 kg/ha magnesium sulfate (25% Mg). These fertilizer quantities could complementarily participate in soil fertility, and further improve plant growth and yields.
4. However, the low interactive contribution to the soil in terms of heavy metals did cause a minimum increase in their metal concentration in both the fescue and lettuce soils, indicating low pollution indices, which suggested very low to limited soil toxicity.
5. It was concluded that the elemental interactions could be a helpful method for the quantitative evaluation of metals and nutrients accumulating in soil under the reuse of wastewater and biosolids by the elemental interactions.

## 5. Highlights

1. Elemental interactions can provide a useful tool for the evaluation of the interactive contribution to soil fertility in terms of heavy metals.
2. The interactions may provide or remove nutrients and heavy metals from/to soil depending on their interactive orientation (synergistic or competitive).
3. They contribute positive or negative changes to soil's physical and chemical properties and affect its functions.
4. They may also determine, to a significant extent, the fertility and toxicity of the soil by adding to soil or removing heavy metals from it, respectively.

**Author Contributions:** Conceptualization, P.H.K., S.S.K. and I.K.K.; methodology, P.H.K., P.K., S.S.K. and I.K.K.; software, S.S.K.; investigation, P.K.; data curation, P.H.K. and S.S.K.; writing—original draft, P.H.K. and S.S.K.; writing—review and editing, P.H.K., P.K. and S.S.K.; resources, G.N.; supervision, I.K.K. All authors have read and agreed to the published version of the manuscript.

**Funding:** This research received no external funding.

**Data Availability Statement:** Data is available upon request.

**Conflicts of Interest:** There is no conflict of interests between the authors or with any other person or agent.

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
