# Peer review of "The Impact of the Elemental Interactions on Soil Fertility and Toxicity in the Presence of Wastewater and Biosolids: A Quantitative Evaluation"

_water, doi:10.3390/w15213743_

Round 1
Reviewer 1 Report (Previous Reviewer 3)
Comments and Suggestions for Authors
The authors revised the manuscript based on the remarks to improve the manuscript for the reader’s understanding.
Accept.
Author Response
Thank you very much for taking the time to review this manuscript.
Reviewer 2 Report (New Reviewer)
Comments and Suggestions for Authors
The paper is dealing with the quantitative evaluation of the elemental interactions in soil irrigated and amended by wastewater and biosolids.
Despite the importance of the research topics and the need to better understand the interaction between chemical elements and their impacts on soil, however the paper is not well presented.
The abstract is not clear, and the introduction does not reflect the paper problematic and objective.
The introduction is not coherent and contains a subtitle "1.1 The quantitative contribution of elemental interactions" that could be avoided.
The presentation and interpretation of the results are not well comprehensive.
Comments on the Quality of English Language
Extensive editing of English language required.
Author Response
Thank you very much for taking the time to review this manuscript. The attached file contains a point-by-point response to your suggestions and comments.
We hope that you will find the new version of our manuscript compatible with the requirements of the Water Journal.

Round 2
Reviewer 2 Report (New Reviewer)
Comments and Suggestions for Authors
The authors have improved the paper.
This manuscript is a resubmission of an earlier submission. The following is a list of the peer review reports and author responses from that submission.
Round 1
Reviewer 1 Report
Comments and Suggestions for Authors
I have the following concerns for the authors:
Title:
The title is a bit misleading. It should be precise and standing alone.
Abstract:
The abstract needs significant revision. It is difficult to read through and grasp points. The results are not indicated.
Introduction:
Although it is well described using a few relevant literature, the objective of the study lacks clarity. The objective better be rewritten in ways meeting the topic. There is a long list with redundant ideas.
Materials and Methods:
Lacks organization: the treatment of the experiments and research design are not clear.
Results and Discussion:
Results are presented but not adequately discussed. They should be discussed and compared and contrasted with findings from previous research.
Reviewer 2 Report
Comments and Suggestions for Authors
The manuscript has many shortcomings and must be rejected in its present form. The biggest concern is to use data from Ntzala (18), who is not a co-author. Detailed notes: The abstract should also briefly present the results and conclusions.
The methodology lacks information on the composition of the sewage used, how it was dosed on the experimental plot, how the fescue was planted. What methods were used to determine 6 nutrients and 9 metals.
Results: Line 111-138 - no results, this description may be in the methods, but it contains unnecessarily general information about regression. Why didn't the authors use multiple regression? With backward regression, they were able to eliminate some elements. Authors should use the coefficient of determination R2, not R.
Line 139 - 156 - this is the methodology. Line 223 - what kind of "sludge" is this? There should be three significant numbers in the equations. Table 2 - these are Ntzal's results. It is not listed when using its data with the table. This data and point 3.5 should be deleted or clarified.
There is no discussion of the results. The conclusion point is rather a summary. You cannot write in the application that the impact is generally positive or negative. What influences the accumulation of metals in the soil?
Please use the multiplication sign, not *.
There are many punctuation errors in the manuscript.
Reviewer 3 Report
Comments and Suggestions for Authors
The manuscript describes the elemental interactions for the evaluation of the contribution to soil fertility and toxicity which is a good topic and falls in the topic of the journal, however, there are some issues to be addressed. The comments are listed below:
1. The English of the text should be checked
2. The authors must include new, relevant, and more information about other studies. Diverse studies are growing attention for diverse uses for pollutant toxicity and removal as reported by the Awual group according to ScienceDirect. The authors need to indicate such points for a broad range of readers. Moreover, the authors need to cite high-impact articles to make the manuscript high-level. The following specific articles should be noted in the revision stage of https://doi.org/10.1016/j.molliq.2023.122854; https://doi.org/10.1016/j.surfin.2023.103276; https://doi.org/10.1016/j.molliq.2023.122763
3. Comparison between the obtained results and those measured in this study with other reported studies should be done and included for more clarity (indicate values not just the number of references).
4. A schematic mechanism describing the adsorption process must be indicated and included (reactions, interactions, etc.)
5. Correct the References using the guide of the Journal. More Conclusions must be included with the best results, and values obtained.
Comments on the Quality of English LanguageThe English language needs to be checked carefully in the revision stage because of many careless mistakes in many positions.